# Phonon behavior in a random solid solution: a lattice dynamics study on the high-entropy alloy FeCoCrMnNi

Shelby R. Turner [1,2,3], Stéphane Pailhès[3], Frédéric Bourdarot [4], Jacques Ollivier [1], Yvan Sidis [5], John-Paul Castellan[5,6], Jean-Marc Zanotti[5], Quentin Berrod [7], Florence Porcher[5], Alexei Bosak[8], Michael Feuerbacher [9], Helmut Schober[1,10], Marc de Boissieu[2] & Valentina M. Giordano [3] ✉

High-Entropy Alloys (HEAs) are a new family of crystalline random alloys with four or more elements in a simple unit cell, at the forefront of materials research for their exceptional mechanical properties. Their strong chemical disorder leads to mass and force-constant fluctuations which are expected to strongly reduce phonon lifetime, responsible for thermal transport, similarly to glasses. Still, the long range order would associate HEAs to crystals with a complex disordered unit cell. These two families of materials, however, exhibit very different phonon dynamics, still leading to similar thermal properties. The question arises on the positioning of HEAs in this context. Here we present an exhaustive experimental investigation of the lattice dynamics in a HEA, $Fe_{20}Co_{20}Cr_{20}Mn_{20}Ni_{20}$, using inelastic neutron and X-ray scattering. We demonstrate that HEAs present unique phonon dynamics at the frontier between fully disordered and ordered materials, characterized by long-propagating acoustic phonons in the whole Brillouin zone.

Recently, a new family of crystalline metallic materials has been discovered and has come to the forefront of materials research for their exceptional mechanical properties: High-Entropy Alloys (HEAs)[1–4]. Obtained with casting techniques from the melt, HEAs are single-phase, equiatomic alloys with four or more elements that are evenly dispersed in an average ordered, close-packed, and simple crystalline structure, forming a random solid solution[5,6]. The strong chemical disorder within the unit cell introduces disorder at a larger lengthscale, as the periodically repeated unit cell is in fact always different due to the random atomic arrangement within it. As such, the translational invariance is disrupted, drawing these materials closer to glasses. Indeed, HEAs share with these latter the presence of a distribution in

atomic sizes, masses and force-constants, usually absent or present only in a limited extent in crystalline materials. Moreover, HEAs also exhibit thermal transport properties quite similar to glasses: thermal conductivities that are much lower than in simple metals, going from some tens of W/mK[7] down to less than 2 W/mK[8], and a low and almost temperature independent lattice contribution[9,10]. In glasses, the emergence of a low and weakly temperature dependent lattice thermal conductivity has been ascribed to a strong phonon scattering due to the intrinsic disorder, which includes topological, mass and force-constant disorder[11–13]. While there are indications that force-constant disorder at the nanoscale is mainly responsible for such strong scattering[13–15], it is experimentally impossible to separate the effect of

[1]Institut Laue-Langevin, F-38042 Grenoble, France. [2]Université Grenoble Alpes, CNRS, Grenoble-INP, SIMaP, F-38000 Grenoble, France. [3]Institute of Light and Matter, UMR5306 Université Lyon 1-CNRS, Université de Lyon, F-69622 Villeurbanne, France. [4]Université Grenoble Alpes, CEA, IRIG, MEM, MDN, F-38000 Grenoble, France. [5]Université Paris-Saclay, CNRS, CEA, Laboratoire Léon Brillouin, F-91191 Gif-sur-Yvette, France. [6]Institut für Festkörperphysik, Karlsruher Institut für Technologie, D-76021 Karlsruhe, Germany. [7]Université Grenoble Alpes, CEA, CNRS, IRIG-SyMMES, F-38000 Grenoble, France. [8]European Synchrotron Radiation Facility, F-38043 Grenoble, France. [9]Peter Grünberg Institut PGI-5 and ER-C, FZ Jülich GmbH, D-52425 Jülich, Germany. [10]European Spallation Source, ERIC, P.O. Box 176, SE-221 00 Lund, Sweden. ✉e-mail: valentina.giordano@univ-lyon1.fr

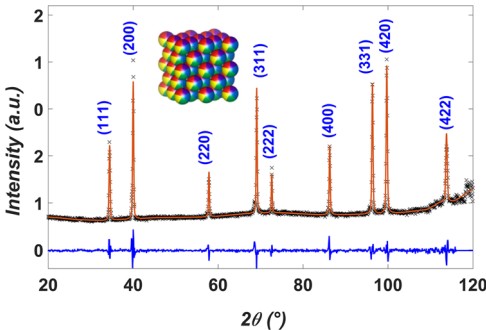

**Fig. 1 | Neutron diffraction.** The neutron diffraction pattern of the polycrystalline sample of FeCoCrMnNi, used for neutron experiments, is reported, measured at 3T-2@LLB with an incident neutron wavelength of 1.230 Å at 300 K. Peaks have been fitted with LeBail-type pattern matching with a FCC structure, space group $Fm\bar{3}m$. The fit is reported as a solid red line, observed data points in black and difference plot in blue. The inset is a visual interpretation of the FeCoCrMnNi FCC structure (made with VESTA[76]) in which each of the 5 elements has equal chance of occupying each atomic position, creating a random solid solution.

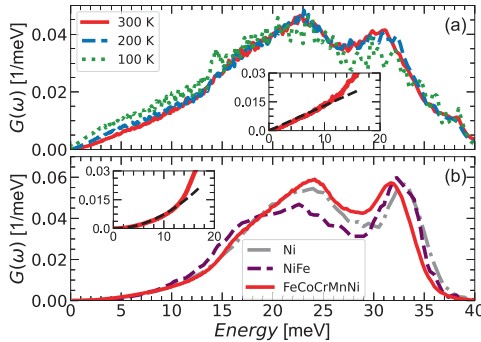

**Fig. 2 | Generalized Vibrational Density of States (GVDOS) of single-crystal and polycrystalline samples of the HEA FeCoCrMnNi. a** GVDOS measured on the polycrystalline sample at IN6-SHARP@ILL with an incident neutron wavelength $\lambda$ = 5.1 Å at three temperatures, resulting in a **Q** range of 0-2.1 Å$^{-1}$ at $S(\mathbf{Q}, E = 0)$. **b** GVDOS measured on the single crystalline sample on IN5@ILL with a neutron wavelength $\lambda$ = 3.2 Å, resulting in a **Q** range of 0-3.6 Å$^{-1}$ at $S(\mathbf{Q}, E = 0)$, compared to that of the single element Ni[46] and the binary alloy NiFe[46]. Insets report the linear and squared dependencies (fits in black dashed lines) of FeCoCrMnNi in **a** and **b** at 300 K, respectively.

the different kinds of disorder in glasses, as they are entangled. The situation is different in random alloys, where the topological disorder is absent and is replaced only by a local strain due to the atomic size disorder. The effect of mass fluctuations on phonon dynamics has been largely investigated in binary alloys both experimentally and theoretically[16–18], while the effect of force-constant fluctuations, experimentally evidenced in a few binary alloys[19,20], has been more challenging to calculate. As such, it has been explored theoretically and experimentally only recently in binary and ternary alloys[21,22].

In this context, HEAs offer a unique playground for tuning the relative weight of these different features and for isolating one of them, thanks to the possibility of customizing their composition[22,23]. As such, the study of their phonon dynamics promises to shed light on the role of force-constant versus topological and mass disorder in determining the strong scattering regime and the origin of the low and weakly temperature dependent thermal conductivity. Interestingly, similar thermal transport properties have also been reported in some crystalline materials[12,24–27] belonging to the family of the Complex Metallic Alloys (CMAs), described in average with a simple cubic cell containing a large number of atoms organized in local (sub)nanostructures and which are frequently associated with intrinsic disorder[25,28]. In this case, the microscopic origin of the glass-like thermal properties has been identified in a very different mechanism with respect to glasses: not a strong phonon attenuation, but a significant reduction of the acoustic phonon spectrum due to the presence of a large number of optic modes[29–34].

The question then arises on the position of HEAs in this context, as they partially share structural properties with both glasses and CMAs, which however are characterized by quite different phonon dynamics. There are two possibilities, which remain elusive at this day, due to the current lack of studies of their vibrational properties: either they can be associated to one of these families, or they may represent a new family of materials with its own specific dynamics.

The aim of this paper is to answer this question by thoroughly investigating the phonon dynamics in a prototype HEA, and comparing its vibrational properties to the ones which have been dubbed as responsible for similar thermal behavior in glasses and CMAs.

Here we provide the first experimental investigation of individual phonon properties in a HEA: Fe$_{20}$Co$_{20}$Cr$_{20}$Mn$_{20}$Ni$_{20}$ (FeCoCrMnNi)[35]. Built from 5 direct-neighbor elements in the periodic table, FeCoCrMnNi exhibits large differences in neither atomic mass nor atomic size[36]. As we will show, this HEA presents phonon dispersions which closely match those of the simple elements composing it. However, phonon lifetimes are much shorter, and can be understood

in terms of scattering from force-constant fluctuations, as predicted by the most recent theories[22], similarly to what happens in glasses. Still, phonons remain well defined and propagate over the entire Brillouin zone, losing their propagative character only at a lengthscale comparable with the nearest neighbors distance. As opposed to both glasses and CMAs, the propagative acoustic phase space extends up to energies of ~20–30 meV and wavevectors larger than 1 Å$^{-1}$, pointing to HEAs as a new class of materials, with unique phonon dynamics, at the frontier between ordered and fully disordered materials.

## Results

### Sample characterization

FeCoCrMnNi is known to be a single-phase alloy of FCC structure[2,37]. In this work we used both polycrystalline and single-crystal samples, prepared as detailed in Section IV, and with an overall composition respectively of Fe$_{19.89}$Co$_{20.97}$Cr$_{17.82}$Mn$_{19.54}$Ni$_{21.78}$ at.% and Fe$_{20.00}$Co$_{19.64}$Cr$_{20.33}$Mn$_{20.10}$Ni$_{19.94}$ at.% (see Supplementary Note 1).

The polycrystalline sample was investigated by neutron diffraction at the Laboratoire Léon Brillouin (LLB) using the thermal-neutron two-axis powder diffractometer 3T-2@LLB with an incident wavelength $\lambda$ = 1.230 Å at 300 K. The diffraction pattern is shown in Fig. 1. A pattern matching (Le Bail fit) was applied to refine the Bragg peaks according to an FCC structure within the $Fm\bar{3}m$ space group with lattice parameter a 3.601(1) Å. More details are given in Supplementary Note 1.

Possible short range ordering and strain distribution were investigated by neutron and X-ray diffuse scattering and are reported in Supplementary Note 1.

### Generalized vibrational density of states

Figure 2(a) reports the Generalized Vibrational Density of States (GVDOS) of a polycrystalline sample after normalization by the Bose-Einstein temperature dependence, as measured at 100, 200, and 300 K. The most striking feature is the lack of a clear acoustic regime at low energy: rather than the usual Debye-like $(\hbar\omega)^2$ behavior, we observe a linear dependence up to ~12 meV. Moreover, the signal in this energy range does not follow the Bose-Einstein dependence on temperature, increasing in intensity with decreasing temperature, and we observe the appearance of a small additional peak at 14 meV at 100 K. This behavior is in fact due to a dominant magnetic scattering signal also detected when measuring at low **Q**. Indeed, FeCoCrMnNi is known to be magnetic and exhibits a magnetic transition at low

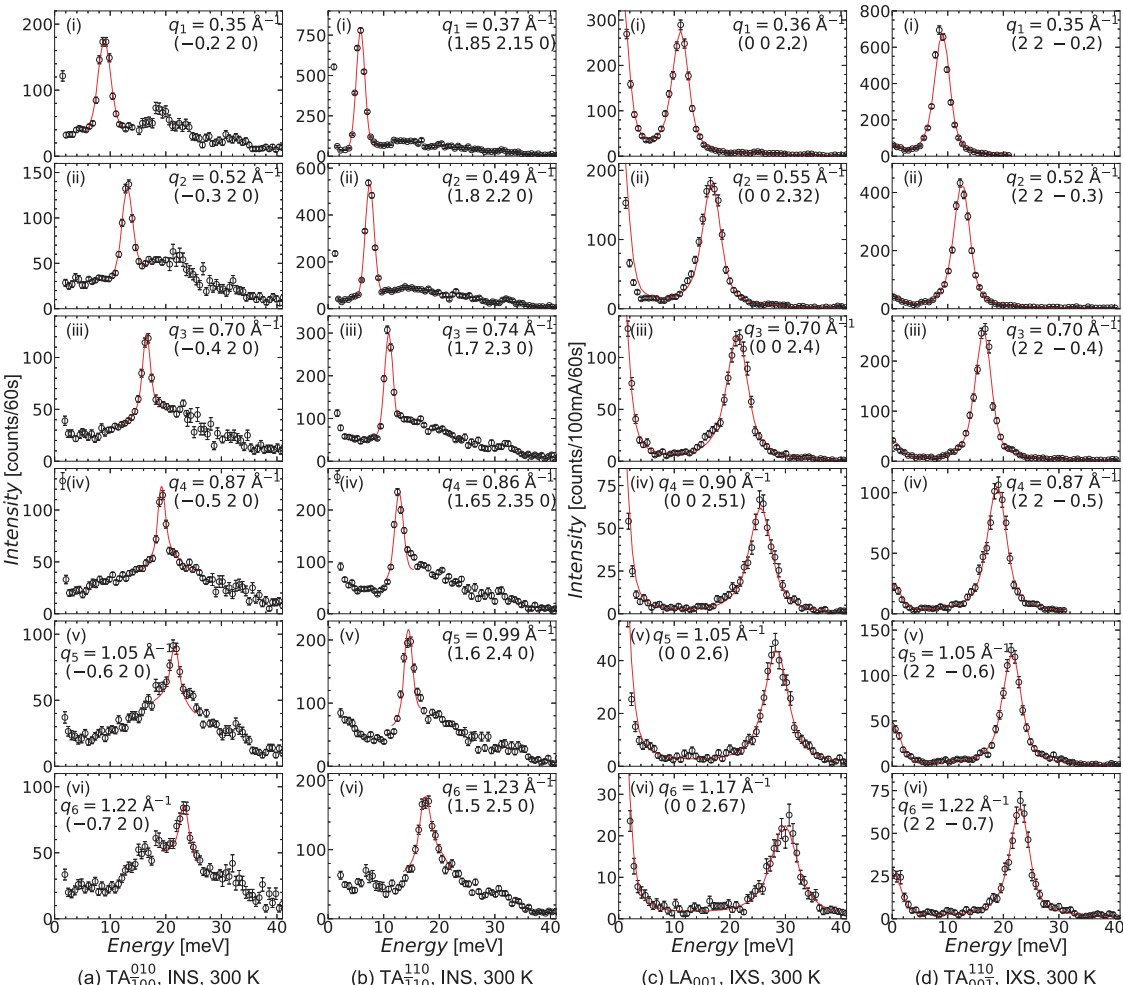

**Fig. 3 | Longitudinal and transverse acoustic (LA,TA) phonon energy scans measured at 300 K. a** The TA dispersion propagating along [$\bar{1}$00], polarized along [010], and **b** the TA dispersion propagating along [$\bar{1}$10], polarized along [110]. **c** The LA dispersion propagating along [001], and **d** the TA dispersion propagating along [00$\bar{1}$], polarized along [110]. Subplots in **a**, **b** have been measured by inelastic neutron scattering (INS), and those in (c,d) have been measured by inelastic X-ray scattering (IXS). All phonon modes were fit as damped harmonic oscillators (solid red lines). IXS scans represent different detectors on the instrument ID28@ESRF, and intensities have not been scaled for efficiencies between detectors. (This is also the reason some subplots have small [$\zeta\zeta$0] components.) The phonon wavevector has been matched across the different polarizations as best as possible. Vertical error bars are s.d. arising from counting statistics.

temperature[9,38–43]. A full interpretation of magnetism in FeCoCrMnNi is outside the scope of this article. Still, magnetic fluctuations could also play a role in enhancing phonon scattering[23,44].

In order to reduce the magnetic contamination, the measurements were performed with a smaller neutron wavelength on a single crystal at room temperature. Here, as specified in Section IV, the data collection encompasses a larger **Q** range, minimizing the relative weight of the low **Q** magnetic scattering in the average. Specifically the intensity of the magnetic contribution is reduced by a factor of ~75%. As shown in Fig. 2(b), we now recapture a squared dependence of the acoustic region between 0 and 12 meV, confirming the Debye-like behavior of acoustic phonons in this HEA, in agreement with previous calculations on random alloys[45]. This GVDOS compares well to those of Ni and NiFe[46], also reported in the figure, as well as to other binary alloys of the same elements[46–49] and the HEA FeCoCrNi[43,50], reported in Supplementary Note 4. This can be explained by the fact that the neutron coherent scattering cross section of Ni and Fe are very similar to each other and much larger than the ones of the other elements, fully dominating the signal in all these materials.

### Individual acoustic phonon properties

We have successively investigated the longitudinal and transverse acoustic (LA,TA) phonon dispersions. Figure 3 reports some selected

spectra measured at 300 K of four polarizations: the TA mode propagating along [$\bar{1}$00], polarized along [010] (TA$_{\bar{1}00}^{010}$), and the TA mode propagating along [$\bar{1}$10], polarized along [110] (TA$_{\bar{1}10}^{110}$) (Fig. 3(a, b)), measured by inelastic neutron scattering (INS) and the LA mode propagating along the [001] direction (LA$_{001}$) and the TA mode propagating along [00$\bar{1}$], polarized along [110] (TA$_{00\bar{1}}^{110}$) (Fig. 3(c, d)), measured by inelastic x-ray scattering (IXS). The measurements were performed around the intense (002) and (220) Bragg peaks. We remind the reader here that, given the cubic symmetry, TA$_{00\bar{1}}^{110}$ and TA$_{\bar{1}00}^{010}$ correspond to the same propagation direction but different polarizations. Additional scans of each of these polarizations, and those of the LA mode propagating along the [110] direction (LA$_{110}$) and the TA mode propagating along [110], polarized along [001] (TA$_{110}^{001}$), are reported in Supplementary Note 5.

Surprisingly for a system with such a strong chemical disorder, a well-defined phonon peak (i.e. $\Gamma \ll \hbar\omega$) propagates in all polarizations until 25–30 meV, at the Brillouin zone boundary. In the INS scans, the phonon disperses on top of a broad, textured band in energy, which is roughly constant at all $q$ within a given direction (see Supplementary Fig. S8), but appears to have slightly different texture and shape in each direction while still exhibiting two major features centered at 20 and 30 meV (see Fig. 3(a.ii) and (b.iii) for example). This feature can be understood as incoherent neutron scattering, which, in this sample,

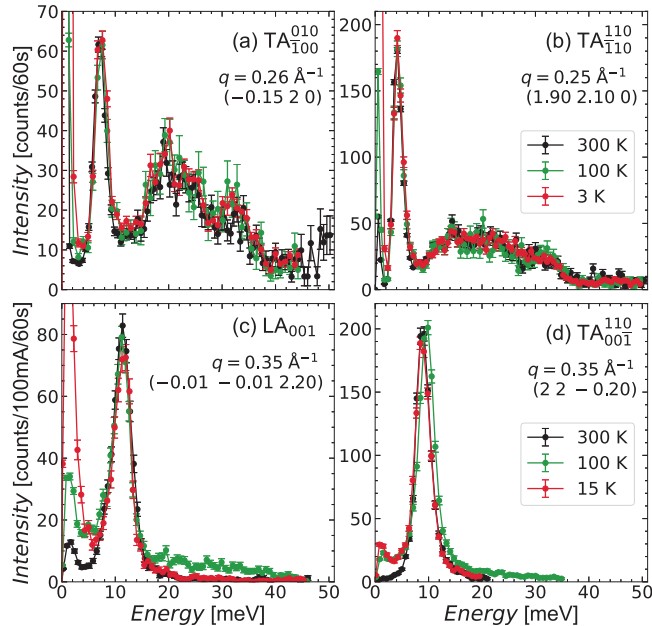

**Fig. 4 | Temperature dependence of representative energy scans across the acoustic phonon dispersions shown in Fig. 3.** Subplots (**a**, **b**) have been measured by inelastic neutron scattering, and (**c**, **d**) by inelastic X-ray scattering. The mode polarization and wavevector are indicated in each subplot. Vertical error bars are s.d. arising from counting statistics.

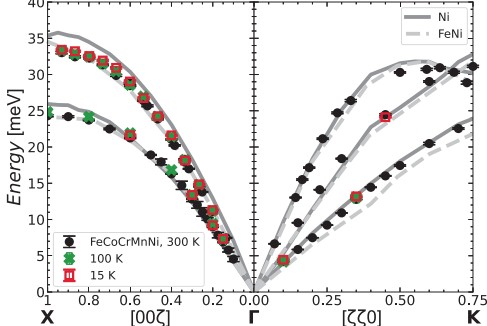

**Fig. 5 | Longitudinal and transverse phonon dispersions.** The energies of the longitudinal and transverse acoustic phonons of FeCoCrMnNi measured in the high symmetry directions [00$\zeta$] and [$\zeta\zeta$0] seen in Fig. 3 have been fit (details in Supplementary Note 2) and plotted at three temperatures against those of the single element Ni[52] and equimolar NiFe[53] measured at 300 K. Error bars correspond to the standard deviation of the fitted parameters.

### Table 1 | Sound velocities

|  | [00$\zeta$] | [$\zeta\zeta$0] |
|---|---|---|
| LA | 4.7(1) | 5.7(1) |
|  | 5.034[a] | 5.834[a] |
| TA$_1$ | 3.8(2) | 3.8(2) |
|  | 3.681[a] | 3.681[a] |
| TA$_2$ | 3.8(2) | 2.3(2) |
|  | 3.681[a] | 2.181[a] |

The LA$_{001}$, degenerate branches TA$_{1\overline{1}0}^{110}$ and TA$_{\overline{1}00}^{010}$, LA$_{110}$, TA$_{00\overline{1}}^{110}$, and TA$_{1\overline{1}0}^{110}$ mode sound velocities of FeCoCrMnNi are given in the table above. In the [$\zeta\zeta$0] direction, TA$_1$ refers to the TA$_{00\overline{1}}^{110}$ branch, and TA$_2$ to the TA$_{1\overline{1}0}^{110}$ mode. They are compared to the sound velocities calculated using the experimentally measured single-crystal elastic constants from Wu et al.[51], marked by [a]. Velocities are written in km/s.

should account for about 70% of the total signal, as predicted by a simple elemental neutron scattering cross-section calculation, including the Laue contribution due to the random atomic distribution. Moreover, as a first approximation, it should be proportional to the density of states multiplied by the square of the energy, and, using our measured GVDOS, the simulated incoherent scattering agrees well with the experimental signal, as shown in Supplementary Note 2. Our interpretation is also confirmed by the fact that the textured band is absent in the IXS measurements, where the X-ray incoherent contribution, related to the different atomic scattering factors of the elements, is negligible in FeCoCrMnNi, thanks to their similar atomic numbers.

The phonons' temperature dependence is reported in Fig. 4, after normalization by the expected Bose-Einstein temperature dependence. It can be seen that, in the INS experiment between 3 and 300 K, both acoustic phonons and the broad band respect the Bose-Einstein dependence and show no evolution in their energy position and shape with temperature, within our instrumental resolution. Concerning the IXS data measured between 15 and 300 K, the difficulties in the sample alignment at each temperature do not allow us to quantitatively compare the intensities at different temperatures. Still, we can confirm the independence of the phonons' energy positions and shape within our instrumental resolution.

Next, we have fit the data to extract phonon energies, normalized intensity, and linewidths, modeling the coherent inelastic signal with a damped harmonic oscillator (DHO) convoluted with the instrumental resolution (see Supplementary Note 2B, C). First, as explained in Supplementary Note 3, we can confirm the acoustic nature of the observed phonon mode, as it keeps a constant normalized intensity over almost all the Brillouin zone. Second, we report in Fig. 5 the LA and TA dispersions for the [00$\zeta$] and [$\zeta\zeta$0] directions at different temperatures. The sound velocities extracted from their low $q$ behavior are in good agreement with the ones calculated from the experimental elastic constants reported in literature on the same material[51], as can be seen in Table 1.

Also reported in Fig. 5 are the literature dispersions of pure Ni[52] and NiFe[53]. The three materials have similar acoustic dispersions in each direction. We have also verified the global agreement with the acoustic dispersions of the other elements and binary alloys composing our HEA, not reported here. This can be understood in terms of the similarity of the sound velocities, related to force-constants and density, of the component elements, as pointed out by Körmann et al.[54].

We now turn our attention to the phonon linewidth, $\Gamma_{FWHM}$, which is directly related to phonon lifetime, $\tau = 2\hbar/\Gamma_{FWHM}$ and mean free path $l = v\tau$. Unfortunately, we could reliably extract it only from X-ray inelastic scattering data, as the strong incoherent scattering entangled with the acoustic phonon in neutron data made the fitting procedure challenging in that case. Details on its extraction in IXS experiments are given in Supplementary Note 2C. In Fig. 6 we report the phonon linewidths at 300 K of the LA$_{001}$ and TA$_{00\overline{1}}^{110}$ polarizations in the [$\zeta$00] direction (panels a and b), and of LA$_{110}$ and TA$_{110}^{001}$ polarizations in the [$\zeta\zeta$0] direction, (panels c and d). Interestingly, the $q$ dependence of phonon broadening appears to be anisotropic and polarization-dependent: longitudinal modes exhibit several regimes, while transverse modes have a smoother dependence with a monotonic increase.

## Discussion

The first question to address is about the origin of the observed phonon broadening. We can rule out a dominant role of anharmonicity, as well as scattering from magnetic fluctuations. Indeed, we could not observe any sizable temperature dependence in either phonon position and/or shape between 300 and 15 K, temperatures which are respectively well above and below the magnetic transition reported at

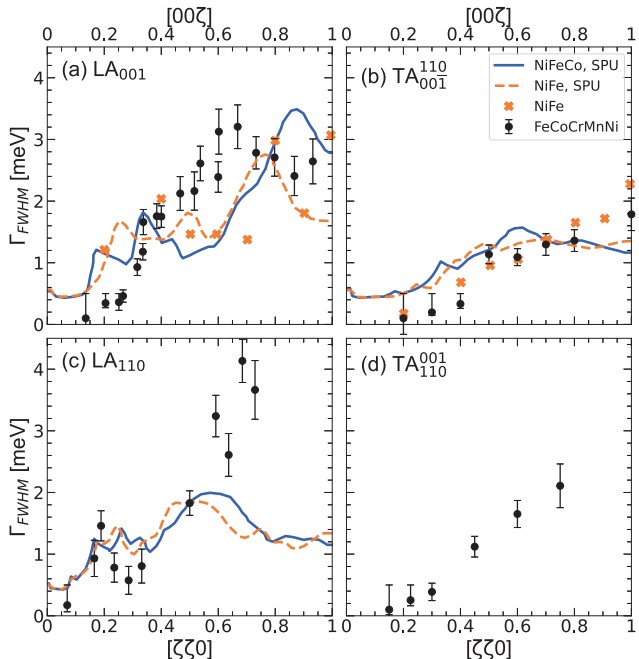

**Fig. 6 | Phonon Linewidths at room temperature.** Intrinsic phonon linewidths of FeCoCrMnNi, measured by inelastic X-ray scattering at 300 K for (a) LA$_{001}$, (b) TA$_{00\bar{1}}^{110}$, (c) LA$_{110}$, and (d) TA$_{100}^{001}$. Experimental data on binary NiFe together with SPU calculations predictions for NiFe and NiFeCo linewidths after Mu et al.[22] are also reported. (SPU calculations were not made for the FeCoCrMnNi polarization shown in (d)). Error bars correspond to the standard deviation of the fitted parameters.

25 K for our HEA[9,38,39]. The observed broadening should therefore be related to the disorder-induced scattering sources associated with the random alloy nature of our HEA. As such, the enhancement of phonon broadening with respect to the simple-element monatomic materials is expected to reflect the role of disorder.

Unfortunately, there is very little literature available on intrinsic phonon linewidths in the elements constituting our FeCoCrMnNi. In the case of Cr[55], measured by IXS with the same energy resolution as our study, the broadening remained resolution limited, confirming thus that the disorder intrinsic to HEAs does increase phonon broadening, which now lies within the experimental resolution.

Surprisingly, despite the large number of elements in our HEA, the phonon linewidths remain similar to the ones reported in random binary alloys of the same elements, such as FeCo and FeNi, this latter also reported in Fig. 6[22].

The difficulty of modeling phonon linewidths in random alloys comes from the complexity of taking into account the quoted three ingredients, i.e. atomic size, mass, and force-constant fluctuations. A simple analytical model, successfully used in mass disordered alloys[56], assumes phonon scattering from isolated defects and leads to the expression $\Gamma_{FWHM} = \pi/2(\hbar\omega)^2 g(\hbar\omega)\langle\epsilon^2\rangle$, where $g(\hbar\omega)$ is the GVDOS and $\epsilon$ represents the sum of all kinds of fluctuations contributions[57]. Using this model, we find that the expected broadening from mass fluctuations in our HEA is negligibly small, at most 0.2 meV. Ascribing the whole observed linewidth to force-constant fluctuations, the global behavior is reproduced quite well along the [$\zeta$00] direction using an average force-constant fluctuation $\epsilon = \langle\Delta F_{ij}/F_{ij}\rangle \sim 20\%$ (see Fig. 7). However, the agreement does not hold in all directions: the observed anisotropy then calls for more complex models.

We therefore turn to the recent theoretical work of Mu et al.[22]. Using the ab initio supercell phonon-unfolding simulations method, the authors demonstrate that, in weak mass disorder alloys such as FeNi, FeCo, and FeNiCo, which are made out of the same constituents of our FeCoCrMnNi, the force-constant fluctuations play a determining

role, being enhanced by the random environment of the atomic pairs. This leads to a significant phonon broadening on the order of 1–2 meV and mainly above $q = 0.7$ r.l.u. as observed both in IXS experiments and calculations, which are also reported in Fig. 6. Unfortunately they did not calculate the same polarization for the transverse mode along the [$\zeta\zeta$0] direction as we have measured, limiting a direct comparison. The behavior looks similar in these materials and our HEA, with different regimes in longitudinal broadening and a smoother behavior in transverse broadening. One should note however that the low-$q$ part of the spectrum could not be reliably reproduced by the simulation, due to the rather small supercell used. Interestingly, the force-constant fluctuations per atomic pair found from the simulation is between 20 and 50%, in nice agreement with our simplistic estimation for FeCoCrMnNi. This similarity between our 5-elements random alloy and binary and ternary alloys highlights a fundamental contradiction between our data and Mu's theory, from which, going from 2 to 5 atoms, one would expect a larger phonon attenuation due to the larger variability of the random environments of the atomic pairs. On the other hand, our results explain the similar thermal behavior recently reported in FeNi and FeCoCrMnNi: if the total thermal conductivity is much larger in the binary than the multi-element alloy, leading to a room temperature value of 300 W/mK in the former against only ~10 W/mK in the latter, once the electronic contribution is subtracted, very similar phonon contributions are found both at 50 and 300 K, of about 4 and 6.9 W/mk respectively[9,58].

As expected, the more advanced theoretical modeling of Mu et al.[22] presents an anisotropy as well, specifically between TA$_{00\bar{1}}^{110}$ and TA$_{110}^{1\bar{1}0}$, confirming that it is intrinsic to the random alloys, and is not signature of a local chemical short range order. Finally, in the light of a dominant force-constant disorder, the observed polarization dependence shares some similarities with the theoretical work of Overy et al., which predicts in this case a stronger effect on longitudinal modes[59]. The opposite has been found in molecular dynamic simulations on metallic glasses[60], where, however, force-constant disorder is entangled with topological, mass and atomic size disorder. Still, when reported as a function of energy, as in Fig. 7, broadening looks similar between the two polarizations in the same energy range. More work is needed to definitely assess the role of the different kinds of disorder on phonon attenuation depending on their polarization.

As mentioned in the introduction, it is worth comparing the vibrational dynamics of our HEA to glasses and to crystalline materials with very large unit cells (CMAs), to identify similarities and differences and establish the place of HEAs with respect to these classes of materials. Specifically, as previously said, the low and weakly temperature dependent thermal conductivity in glasses has been associated to the existence of a strong phonon scattering due to the intrinsic disorder. Such regime, characterized by $\Gamma_{FWHM} \propto (\hbar\omega)^4$, has been reported in some glasses at energies corresponding to the deviation of the GVDOS from the Debye behavior and has lately been ascribed mostly to the force-constant disorder at a nanometric lengthscale[13–15]. Interestingly, the same phenomenology arises in random matrix approaches, which model the vibrational properties of a glass through a random network of force-constants over a regular lattice, quite similarly to the case of our HEA[61,62]. It is thus important to check weather the same behavior can be found in our HEA, where force-constant fluctuations are the dominant phonon scattering source.

As we have seen, acoustic attenuation in our HEA is anisotropic, which is not the case in glasses. We thus do not find a universal energy dependence for all polarizations and directions. Still, if we focus on the [$\zeta$00] propagation direction below 30 meV, we observe a crossover from a strong to a weak dependence in the longitudinal polarization, which can be rationalized as two distinct power laws, $\Gamma_{FWHM} \sim (\hbar\omega)^{4.5\pm1}$ up to ~18 meV, and $(\hbar\omega)^{1.7\pm0.2}$ above, as can be seen in Fig. 8, where data are reported in a logarithmic scale.

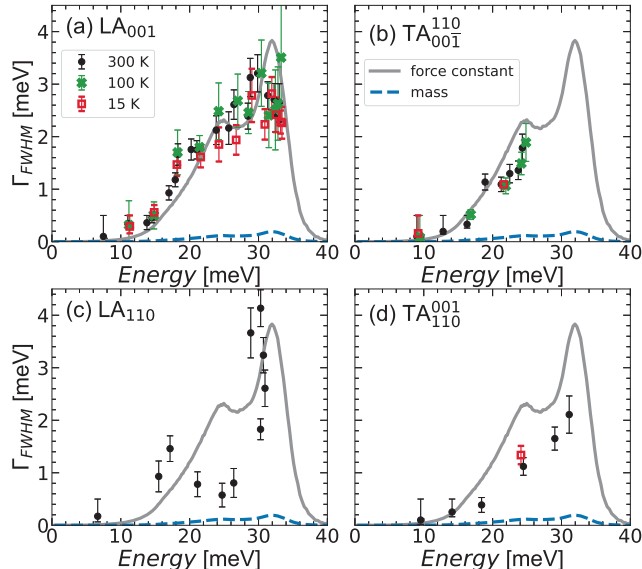

**Fig. 7 | Perturbation theory modeling of phonon linewidths.** Intrinsic phonon linewidths, measured by inelastic X-ray scattering at three temperatures for (**a**) $LA_{001}$, (**b**) $TA_{00\bar{1}}^{110}$, (**c**) $LA_{110}$, and (**d**) $TA_{100}^{001}$, reported as a function of energy and compared with a perturbation theory calculation of the broadening, based on calculated mass fluctuations and estimated force-constant fluctuations.

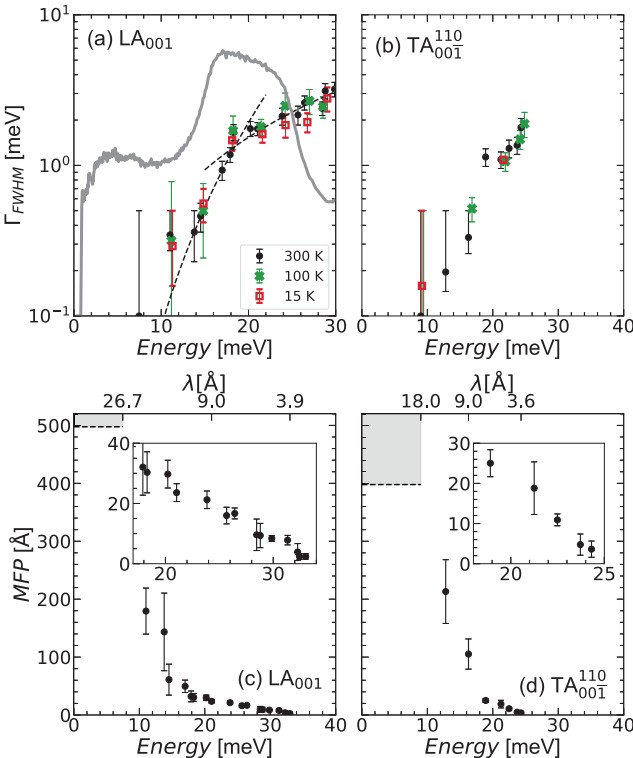

**Fig. 8 | Intrinsic phonon linewidths, measured by inelastic X-ray scattering.** Subplots (**a**, **b**) show linewidths measured at three temperatures. For the sake of clarity, in (a) we report only the first two broadening regimes, up to 30 meV, for the longitudinal mode, together with $(\hbar\omega)^{4.5}$ and $(\hbar\omega)^{1.7}$ lines (black dashed lines) as a guide to the eye. The FeCoCrMnNi generalized vibrational density of states (GVDOS) originally shown in Fig. 2(b), divided by energy squared and scaled, is plotted in gray in (**a**) for reference. Subplots (**c**, **d**) are mean free paths (MFPs), calculated from the intrinsic linewidths at 300 K. Error bars correspond to the standard deviation as calculated using the error propagation theory. Below - 10 meV the phonon linewidth could not be resolved with our experimental resolution. This allows us only to estimate an upper limit for intrinsic linewidth, i.e. a lower limit for lifetime and MFP. The shaded areas in the figure indicate the region in which, on this basis, the real MFP lies.

Here we also see, superposed to the attenuation behavior, the GVDOS divided by energy squared : the strong scattering regime clearly corresponds to the deviation of the GVDOS from the Debye behavior, and the second regime settles in when the reduced GVDOS reaches its maximum at the Van Hove singularity. This correspondence can be understood as due to the fact that, when the phonon starts to see the atomistic details, leading to the deviation from the Debye prediction, it also starts feeling the force-constant disorder, leading to the strong scattering regime. The behavior seems thus similar to glasses, although the lengthscales are different, the strong scattering regime ending at some 20–30 Å in glasses against 10 Å in our HEA. It is however important to underline that the presence of the $(\hbar\omega)^4$ regime is not ubiquitous in glasses. Specifically, in metallic glasses (MGs), contradictory results have been reported, with a milder energy dependence and the sequence $(\hbar\omega)^2-(\hbar\omega)$[63–66]. Interestingly, and quite similarly to the case of our HEA, the possible sequence of regimes $(\hbar\omega)^4 - (\hbar\omega)^2 - (\hbar\omega)$ has also been reported[67].

Another fundamental difference appears: the crossover from a stronger to a milder energy dependence usually corresponds to the Ioffe-Regel limit (IR) in glasses, identified by the condition that the phonon mean free path becomes comparable to its wavelength, $l \sim \lambda$, such that phonons with smaller wavelengths cannot be considered as propagative anymore[68]. This is not the case for FeCoCrMnNi, as can be seen in Fig. 8(c, d), where we report the longitudinal and transverse mean free paths as a function of phonon energy. Focusing on the longitudinal polarization, we see that only for energies above - 30 meV, the mean free path is comparable with the phonon wavelength. The IR crossover in our HEA is thus located at wavelengths comparable with the unit cell size, one order of magnitude smaller than in the majority of glasses.

It is interesting to note that the IR crossover may be found at much smaller lengthscales in some glasses, generally characterized by a minor degree of disorder, such as MGs with an important medium-range order, up to 1–2 nm[69], and monatomic amorphous silicon[70], where force-constant fluctuations are only a consequence of a distribution of bond-lengths[71].

We can thus conclude that, even if we can find in one direction and polarization a strong scattering regime due to force-constant fluctuations, as found in many glasses, phonon dynamics in our HEA remains different as phonons remain propagative in a much larger wavelength and energy range than in glasses with important force-constant disorder.

A similar strong scattering regime ($\Gamma_{FWHM} \propto (\hbar\omega)^4$) has been reported as well in some directions or polarizations in some quasicrystals[72–75], and in the periodic quasicrystal approximant o-Al$_{13}$Co$_4$[30] and has been ascribed to acoustic-optic hybridization or mode mixing[30,73,74]. Still, here as well, a global view of their dynamics establishes major differences between HEAs and CMAs. First, the strong scattering regime is not present in all CMAs. Specifically, it is absent in clathrates, where the lifetimes and mean free paths are also much longer than in quasicrystals and approximants[33]. Still, all CMAs, including clathrates, are characterized by the separation of the phonon spectrum in a reduced acoustic phase space, typically limited to wavevectors smaller than 0.3–0.5 Å$^{-1}$ and energies smaller than 8-10 meV, and a large phase space dominated by dispersionless optic modes. The situation in our HEA is clearly different: first, the continuum of dispersionless excitations is not observed, and second, the acoustic regime extends to much higher energies and wavevectors, up to 22 and 30 meV for TA and LA respectively, close to the Brillouin zone boundary. Moreover, phonon attenuation in our HEA, while larger than in clathrates, is smaller than in quasicrystals.

In conclusion, we have reported the first experimental investigation of the acoustic phonon dispersions in a High-Entropy Alloy, complemented by the GVDOS survey, at room and low temperature. FeCoCrMnNi is a 5-element random alloy, and, despite the inherent disorder, its phonon dynamics closely resemble that of the pure elements composing it and the corresponding binary alloys. More specifically, acoustic phonons disperse throughout the whole Brillouin zone up to energies between 20 and 30 meV, exhibiting an attenuation stronger than in simple elements, but of the same order of magnitude as in binary alloys. The whole attenuation behavior can be understood in terms of scattering from force-constant fluctuations, which have been estimated to amount to ~20% using a simple theoretical model, a value comparable with calculations of Mu and co-authors on binary and ternary alloys made of the same elements. If this is unexpected from Mu's theory, which would predict increased force-constant fluctuations due to the larger variability of random environments of a given atomic pair, it explains recent experimental reports on comparable phonon lattice thermal conductivities in our HEA and FeNi[9]. As such, our findings represent a precious database which may be used as a benchmark for future theoretical predictions, stimulating further theoretical developments for describing polyatomic random alloys.

Finally, the analysis of our results confirms that, while they share some structural aspects and thermal properties with both glasses and CMAs, HEAs exhibit specific phonon dynamics, with marked differences from the ones of those systems: the acoustic regime is preserved across the whole Brillouin zone, while in glasses and CMAs it is reduced in a rather small $(q, \hbar\omega)$ range, except in a few cases of glasses where either the topological or the force-constants disorders are very reduced. Moreover, despite the strong chemical disorder, phonon attenuation remains much smaller than in glasses and quasicrystals, and more similar to that of simple binary alloys. Still, all these materials present a strong phonon scattering regime, $\Gamma_{FWHM} \propto (\hbar\omega)^4$, in at least some polarizations and directions, which arises simultaneously to the deviation of the acoustic dynamics from the Debye behavior.

Surprisingly, our work seems to suggest that the high entropy mixing of many elements is ineffective in terms of phonon engineering, while it can be used to effectively impact electron transport and thus the electronic contribution to thermal transport, as demonstrated by Jin et al.[9]. Still, this result is specific to this HEA, characterized by a force-constant disorder similar to binary alloys. In order to effectively impact phonons, larger force-constants fluctuations should be sought, by choosing elements with more different characteristics. This is the case of HCP and BCC HEAs, where, however, the larger atomic differences introduce as well a stronger mass and atomic size disorder, so that the force-constant disorder does not dominate anymore.

## Methods

### Sample synthesis and characterization

Polycrystalline samples were produced by alloying an equiatomic composition of high-purity elements Fe, Co, Cr, Mn, and Ni in an inductively coupled high-frequency levitation furnace. After several remelting cycles to achieve a high homogeneity, cylindrical ingots were slip-cast into a water-cooled copper mold. Pieces of ~1 cm³ were cut from the ingots and annealed at 1200 °C for 48 h for homogenization.

Single crystals were grown using ingots of ~70 g by means of the Bridgman technique[35]. The pre-alloyed material was fit in a cylindrical, alumina crucible of ~10 cm length and an internal diameter of 20 mm and a 30° tip-shaped bottom, and inserted in a vertical tube furnace. The tip of the crucible was placed on a movable rod equipped with a cold finger to create a defined and steep temperature gradient. The furnace temperature was set to 1340 °C, above the melting temperature of 1330 °C of the alloy, and kept constant. The growth process was carried out under an argon atmosphere of 200 mbar. Solidification was effected by lowering the crucible vertically out of the hot zone of the furnace at a velocity of 50 mm/h.

### Inelastic scattering measurements

The GVDOS was measured using the cold-neutron Time-of-Flight (TOF) technique at the Institut Laue-Langevin (ILL). The measurements on the polycrystalline sample were performed on the IN6-SHARP beamline at 100, 200, and 300 K. The neutron incident wavelength was $\lambda = 5.1$ Å, resulting in a **Q** range of 0–2.1 Å$^{-1}$ at $S(\mathbf{Q}, E = 0)$. The measurements on the single crystal were performed on the IN5 beamline at 300 K. The crystal was aligned in the scattering plane ([100]; [010]), allowing wavevectors of $\mathbf{Q} = \frac{2\pi}{a}(\zeta, \xi, 0)$. The neutron incident wavelength was $\lambda = 3.2$ Å, resulting in a **Q** range of 0–3.6 Å$^{-1}$ at $S(\mathbf{Q}, E = 0)$, and encompassing the first Bragg peak visible in the given scattering plane, (200). Details on the TOF data integration and treatment for the resulting GVDOS plots are in Supplementary Note 2. For the individual phonon dispersions measurements, the INS sample, a cylinder rod 10 cm long and 2 cm wide, was aligned in the scattering plane ([100]; [010]). Measurements were taken on the thermal-neutron Triple-Axis Spectrometer (TAS) 1T-1 at LLB at 3, 100, and 300 K, in a standard cryostat environment, with a fixed $k_f = 2.662$ Å$^{-1}$ near the intense Bragg peaks (020) and (220). The IXS sample, $a \sim 100$ μm length needle chemically etched from the INS one, was aligned in the ([001]; [110]) scattering plane. Measurements were performed on the ID28 beamline at ESRF at 15, 100, and 300 K using a displex closed cycle cryocooler. The incoming X-ray wavelength was 0.697 Å and the energy resolution 2.8 meV. More information about the experiment specifics are reported in Supplementary Note 2.

## Data availability

Processed data are reported in the figures of the main text and in the Supplementary Information. Raw data from INS measurements at the ILL are available at https://doi.ill.fr/10.5291/ILL-DATA.7-01-505 and https://doi.ill.fr/10.5291/ILL-DATA.7-01-493, LLB measurements correspond to proposal 658, and ESRF measurements to proposal hc-4327. Data are made publicly available after an embargo of a few years, or may be accessed earlier on request.

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

## Acknowledgements

S.R.T. acknowledges financial support from the ISP program of the IDEX Université Grenoble Alpes. S.P. and V.M.G. acknowledge support from the Lyon IDEX Scientific Breakthrough program (project IPPON, S.P). V.M.G. acknowledges support from the ANR (project MAPS-ANR-20-CE05-0046, V.G.). M.F. acknowledges support from the German Research foundation (DFG; grant No. FE 571/4 within the priority program SPP2006, M.F.). M.dB. acknowledges the support of the Aperiodic project ANR-18-CE92-0014. This work has been carried out within the European C-MetAC network.

## Author contributions

V.M.G. conceived, directed the project, and supervised the data analysis and interpretation. S.R.T. performed all experiments and data analysis and participated in their interpretation. S.P. and M.dB. participated in the analysis and interpretation. F.B. and H.S. participated in the scientific discussions, J.O., Y.S., J.-P.C., J.-M.Z., Q.B., and F.P. assured the experimental support for neutron experiments, A.B. assured the experimental support for X ray experiments, and M.F. prepared and characterized the samples.

## Competing interests

The authors declare no competing interests.
