## [Peer Review File · Nature Communications]

Phonon behavior in a random solid solution: a lattice dynamics study on the high-entropy alloy FeCoCrMnNiREVIEWER COMMENTS

Reviewer #1 (Remarks to the Author):

The experimental results about lattice dynamics in HEAs, especially in Fe₂₀Co₂₀Cr₂₀Mn₂₀Ni₂₀ single crystal, are intriguing. The authors revealed differences in phonon dynamics of a prototypical HEA between fully disordered and ordered materials using INS and IXS. It was found that long-propagating acoustic phonons exist in the whole Brillouin zone, with phonon dispersion and phonon width closely resembling that of binary alloy combinations. It was suggested that the phonon scattering in HEA could be well understood in terms of the force-constant fluctuations rather than the anharmonicity or magnetic fluctuations. However, there are a few concerns:

- (1) Both the phonon dispersion (Fig.5) and phonon width (Fig.6) in high-entropy alloy FeCoCrMnNi are similar to that of binary (low-entropy) alloy NiFe. It seems that the high entropy induced by the chemical disorder has a more negligible effect on the phonon dynamics. It seems to indicate that HEAs can be excluded in designing functional materials related to phonons. This result is fascinating. Further discussion about the similarities/differences between phonon dynamics in HEA and binary alloy should be given.
- (2) The neutron experiment (ref.71) revealed that transverse and longitudinal phonon respond differently to the topological disorder in metallic glasses. In an earlier theoretical study of the lattice spring model (Phys. Status Solidi 254, 1600586 (2017)), Overy also proved that (i) the mass and force-constant fluctuations give rise to different broadening effects on phonon dynamics and (ii) the broadening of transverse and longitudinal phonon are different. This topic is quite interesting, and I hope the authors give more discussion about the differences between transverse and longitudinal phonon broadening (Fig.7) in HEA.
- (3) As shown in Fig.8a, the authors reported a $\omega_{4.5}$ -to- $\omega_{1.7}$ crossover of the longitudinal phonon width in HEA, similar to the ω_4 -to- ω_2 crossover observed in the glasses. Besides, it was also shown that the reduced GVDOS displays a peak around the crossover energy, quite similar to the boson peak in the glass. As far as I can see, the peak observed in HEA is not related to the boson peak in glasses but is close to the case of van hove singularity in ordered crystals. Direct evidence can be found in Fig.2b in that the single element Ni has almost identical GVDOS with the high-entropy alloy FeCoCrMnNi. Thus, it is reasonable to suspect that the reduced GVDOS of Ni also displays a peak around the same energy. However, since there exists no chemical disorder in single element Ni, the peak here must result from the nonlinear phonon dispersion of van hove singularity. Thus, the correlation between the peak in reduced GVDOS and the crossover energy of longitudinal phonon width is unreliable. The authors should clarify the above difference.
- (4) The neutron diffraction data shown in Fig.1 is used a pattern matching fit, whether it can be refined, such as by Rietveld refinement.
- (5) About the phonon DOS data shown in Fig.2, did the author do the background subtraction? If so, how? How was the elastic peak subtracted? What is the INS energy resolution?
- (6) The magnetic transition was reported at 25 K. Why is the extra DOS in 100K and 200K data shown in Fig.2a from magnetic scattering?
- (7) Talking about the Boson peak in Fig.8a, it is also imperative to clarify how the background and elastic peaks are subtracted because they affect the weak Boson peak feature sensitively.
- (8) The broad peak between 15-40 meV in the INS data is suggested from the incoherent scattering by the authors. The incoherent calculation is based on the assumption that the

sample is a perfect single phase. However, this might be challenging for the INS large sample. I am wondering whether these are the optical phonons. Although these are not shown in the IXS data, however, in some cases, the IXS optical phonon scattering intensity may be too weak to be measured. How can we rule out this possibility? It would be very interesting to check it because fully disordered materials do not have optical phonons while ordered materials have optical phonons.

(9) Between lines 311-314, the authors demonstrated the acoustic nature by a constant normalized intensity over almost all of the Brillouin zone. The logic is not clear.

The manuscript is worth being published after addressing the above concerns.

Reviewer #2 (Remarks to the Author):

In the manuscript, Turner et al. report detailed study of phonon dynamics of a high-entropy alloy, Fe₂₀Co₂₀Cr₂₀Mn₂₀Ni₂₀, using inelastic neutron and X-ray scattering. The results demonstrate that phonons in the HEA remain well defined and propagate over the entire Brillouin zone, which is reminiscent of those in binary alloys and simple elements, but opposite of those in glasses and complex metallic alloy. In the light of the fact that HEAs exhibit quite similar thermal transport properties to glasses and CMAs, the present study indicates a different microscopic mechanism in HEAs than in glasses and CMAs.

While it is not the first experimental investigation of the lattice dynamics in the HEA despite that the authors claim it is [see Yang et al., *Journal of Materials Science & Technology* 99 (2022) 55–60], I think the concrete experimental data, competently executed analysis and new insight into the physics of this system merit publication in *Nature Communications*.

On the other hand, a major issue should be addressed before acceptance, that is lack of comparison between the HEA and the binary alloys, particularly their thermal transport properties and relationships to phonon dynamics. In the manuscript, the authors present that the phonon linewidth of the HEA is broader than the simple-element metals, which explains why HEAs bear much lower thermal conductivities than the simple-element metals, going from some tens of W/mK down to less than 2 W/mK. They also show that the HEA and binary alloys share similar phonon linewidths. According to the above-mentioned logic, the thermal transport properties of HEA should be in the similar level as that of binary alloys. However, to my knowledge, the thermal conductivity of FeNi alloy is larger than 2 W/mK [G. Ya. Khadzhay et al., *Low Temp. Phys.* 46, 939 (2020)].

Besides, several typos written as “an HEA” are found in both the abstract and main text, which should be corrected as “a HEA”.

Reviewer #3 (Remarks to the Author):

The manuscript by Turner et al. reports a comprehensive experimental study of phonon

dynamics in high-entropy alloys (HEAs). The sample is the classic quinary alloy, FeCoCrMnNi, also known as the Cantor alloy. The experiments, based on inelastic neutron scattering and inelastic X-ray scattering measurements, were thorough and covered several temperatures. Both polycrystalline and single-crystal samples were studied. The latter allowed mapping of the phonon dispersion curves and determination of phonon lifetimes, although the precision was kind of poor.

The main conclusion of the paper is that in spite of the complicated chemistry, the phonon behavior of this equiatomic multicomponent alloy is similar to that of Ni and the binary NiFe. This can be seen from the generalized density of states as well as from the phonon dispersion curves mapped with a single-crystal sample. The authors also tried to make point about the phonon peak width which characterizes the phonon life time. The authors demonstrated that the broadening of the phonon peak width could be attributed to fluctuation of force constants. The authors also tried to compare the measured peak widths with calculations by Mu et al., but the quality of data really does not allow any definitive conclusions to be drawn (Fig. 6).

While the paper reported extensive experimental data on phonons in FeCoCrMnNi HEA, the authors fell short of extracting insights on phonon behavior. For example, why would this alloy behave the same as Ni or NiFe? There are lots of first-principles based calculations, which the authors have alluded to in their reference list. The authors could use these calculations to help interpret their experimental data at the microscopic level. Out of these considerations, I am unable to recommend the paper for Nature Communications. Instead, the paper may be better suited for a more specialized journal.

A couple of minor comments/questions

1. I don't quite understand why the authors keep referring to glass. HEA is very different from glass. FeCoCrMnNi is a FCC alloy and by and large behaves like a FCC alloy in many aspects, not just in phonons. In fact, as Fig. 8a shows, there is no Boson Peak in this HEA, which is a ubiquitous feature of glass.
2. P. 3, lines 192-196, what is the point of saying "we could see weak signatures of xxx, but this result needs to be confirmed..."?
3. The massive data sets of phonon scans in Fig. 3 could be condensed or better yet presented in Supplemental Information.
4. Fig. 6 and the subsequent discussion are somewhat disconnected. Fig. 6 was first presented in section C, but without much discussion. Discussion of Fig. 6 came after Fig. 7, the comparison with calculations with force constant fluctuations.

REVIEWER COMMENTS

Reviewer #1 (Remarks to the Author):

The experimental results about lattice dynamics in HEAs, especially in Fe₂₀Co₂₀Cr₂₀Mn₂₀Ni₂₀ single crystal, are intriguing. The authors revealed differences in phonon dynamics of a prototypical HEA between fully disordered and ordered materials using INS and IXS. It was found that long-propagating acoustic phonons exist in the whole Brillouin zone, with phonon dispersion and phonon width closely resembling that of binary alloy combinations. It was suggested that the phonon scattering in HEA could be well understood in terms of the force-constant fluctuations rather than the anharmonicity or magnetic fluctuations. However, there are a few concerns:

(1) Both the phonon dispersion (Fig.5) and phonon width (Fig.6) in high-entropy alloy FeCoCrMnNi are similar to that of binary (low-entropy) alloy NiFe. It seems that the high entropy induced by the chemical disorder has a more negligible effect on the phonon dynamics. It seems to indicate that HEAs can be excluded in designing functional materials related to phonons. This result is fascinating. Further discussion about the similarities/differences between phonon dynamics in HEA and binary alloy should be given.

We thank the Referee for this comment and his/her suggestion. We have added some comments in this sense to the manuscript, specifically pointing to the fact that stronger fluctuations are needed for a visible effect of the higher entropy and then its functionality for materials design. This can be achieved by properly choosing the elements in order to enhance the atomic differences, for example in HCP or BCC HEAs. Still, this means as well enhancing mass and atomic size differences, so that the dynamics won't be dominated by elastic constant fluctuations anymore.

(2) The neutron experiment (ref.71) revealed that transverse and longitudinal phonon respond differently to the topological disorder in metallic glasses. In an earlier theoretical study of the lattice spring model (Phys. Status Solidi 254, 1600586 (2017)), Overy also proved that (i) the mass and force-constant fluctuations give rise to different broadening effects on phonon dynamics and (ii) the broadening of transverse and longitudinal phonon are different. This topic is quite interesting, and I hope the authors give more discussion about the differences between transverse and longitudinal phonon broadening (Fig.7) in HEA.

We thank the Referee for the comment and the suggested reference by Overy, which is indeed very interesting. We note that in Ref 71 experimental data do not allow to separate longitudinal from transverse modes in the observed scattered intensity, and only MD simulations allow to extract the widths and see a larger attenuation in transverse than in longitudinal modes, with a q dependence which mimics the one of the S(q). This result does not seem to be universal. As an example, we have reported an IXS study on a metallic glass in Tlili et al., Acta Mater. 136, 425 (2017), where our analysis, taking into account both a longitudinal and a transverse component, shows that the attenuation is quite similar between the two polarizations. Concerning Overy's work he has shown that elastic constant disorder affects more strongly longitudinal than low energy transverse modes. However for a better comparison with experimental results, the simulations should be ran on larger super-cells, and also FCC lattices.

The stronger q dependence in longitudinal than transverse modes is more similar to what predicted by Overy than what has been calculated in MGs: this is a nice agreement, as we don't have any topological disorder (differently from MG) but we have elastic constant disorder (as discussed in Overy's work). Still, phonon broadening should be looked at as a function of energy rather than q: in

our HEA, longitudinal modes are more strongly broadened than transverse modes only at high energy, where in fact transverse modes do not exist. When compared as a function of energy, as reported in Fig. 7, the broadening looks similar in both polarizations in the same energy range. And indeed, both LA_{001} and TA_{001}^{110} are reproduced by the same perturbation model, only based on the GVDOS. We have added this comment in the manuscript, pointing to the need of further theoretical and experimental works allowing to definitely assess the role of the different kinds of disorder on phonon attenuation depending on their polarization.

(3) As shown in Fig.8a, the authors reported a $\omega_{4.5}$ -to- $\omega_{1.7}$ crossover of the longitudinal phonon width in HEA, similar to the ω_4 -to- ω_2 crossover observed in the glasses. Besides, it was also shown that the reduced GVDOS displays a peak around the crossover energy, quite similar to the boson peak in the glass. As far as I can see, the peak observed in HEA is not related to the boson peak in glasses but is close to the case of van hove singularity in ordered crystals. Direct evidence can be found in Fig.2b in that the single element Ni has almost identical GVDOS with the high-entropy alloy FeCoCrMnNi. Thus, it is reasonable to suspect that the reduced GVDOS of Ni also displays a peak around the same energy. However, since there exists no chemical disorder in single element Ni, the peak here must result from the nonlinear phonon dispersion of van hove singularity. Thus, the correlation between the peak in reduced GVDOS and the crossover energy of longitudinal phonon width is unreliable. The authors should clarify the above difference.

We thank the Referee for his/her comment, which shows us that we have not been clear enough in our manuscript on this point. In fact the maximum in the reduced GVDOS of our HEA is not a Boson Peak (BP), but it is indeed the Van Hove singularity (VH). So we agree with the Referee that the peak results from a flattening of the phonon branches or a deviation from the Debye regime. Such a peak in the GVDOS is known to impact drastically on the phonon dynamics in CMA (T. Tadano et al., Phys. Rev. Lett. 120, 105901 (2018))) leading to the exaltation of multi-phonons scattering processes and thus the anharmonicity, and the decrease of the phonon participation ratio (Pailhès et al. Physical Review Letters, vol. 113, p. 025506 (2014)). The deviation from the Debye prediction also means in general that the continuum approximation doesn't hold anymore, and the phonon sees the discretized nature of the medium. As such, the fact that the change of broadening regime corresponds to such deviation (going towards the VH in crystals or BP in glasses) means that when the phonon sees the atomic details it also starts feeling the force constant fluctuations leading to the ω_4 - ω_2 crossover like in glasses. The lengthscale of the importance of such fluctuations is thus much larger than the unit cell. We have modified the manuscript to make this point clearer.

(4) The neutron diffraction data shown in Fig.1 is used a pattern matching fit, whether it can be refined, such as by Rietveld refinement.

The Rietveld refinement of this phase was already reported in the literature (F. Zhang et al. Materials Research Letters, 6:8, 450-455 (2018), DOI: 10.1080/21663831.2018.1478332). In our case, we could not perform another Rietveld refinement as the sample used for neutron diffraction was not powder but polycrystalline with few grains and a non perfect orientational average.

(5) About the phonon DOS data shown in Fig.2, did the author do the background subtraction? If so, how? How was the elastic peak subtracted? What is the INS energy resolution?

Both the IN6 and IN5 GVDOS calculations were done using Muphacor, which uses the natural time-of-flight step. This means that the energy resolution degrades with increasing energy. Specifically, it remains less than 1 meV until ~20 meV and is as large as ~2 meV at around 40 meV. Concerning the corrections in the extraction of the GVDOS, Muphacor accounts for multiphonon scattering, and we also subtracted empty can measurements from all of these measurements as our background subtraction, along with "calibrating" with vanadium.

The elastic line subtraction is part of the standard Muphacor data treatment, which only asks for input the neutron scattering cross-section. It then separates the coherent and incoherent inelastic scattering components from the scattering function, and subtract the elastic peak.

(6) The magnetic transition was reported at 25 K. Why is the extra DOS in 100K and 200K data shown in Fig.2a from magnetic scattering?

What we show in Fig. 2 is a GVDOS obtained integrating at low Q. As such, it includes the magnetic contribution (higher at low Q following the magnetic form factors while the nuclear intensity evolves as Q^2), which is there at all temperatures in our sample, because of the magnetic moment of the elements constituting it. Our HEA is paramagnetic at room temperature and transforms to a spin glass phase at 25K. However, magnetic fluctuations remain important up to room temperature, as we could confirm in a preliminary neutron diffraction experiment as a function of temperature.

(7) Talking about the Boson peak in Fig.8a, it is also imperative to clarify how the background and elastic peaks are subtracted because they affect the weak Boson peak feature sensitively.

In our HEA there is no BP as discussed above. The maximum in Fig. 6 is only the Van Hove singularity of a normal crystalline phase. The very weak bump at about 4-5meV, is, in our opinion, within the error of the measurement and not to be considered as a real peak. This is why we don't comment on it.

(8) The broad peak between 15-40 meV in the INS data is suggested from the incoherent scattering by the authors. The incoherent calculation is based on the assumption that the sample is a perfect single phase. However, this might be challenging for the INS large sample. I am wondering whether these are the optical phonons. Although these are not shown in the IXS data, however, in some cases, the IXS optical phonon scattering intensity may be too weak to be measured. How can we rule out this possibility? It would be very interesting to check it because fully disordered materials do not have optical phonons while ordered materials have optical phonons.

The IXS cross section depends on the atomic number of the elements of our HEA, which is high. Moreover, the optic modes intensity increases with Q and we were at high enough Bragg peak order, so we would expect to be able to see optic modes in the IXS experiment as well, if they were visible in the neutrons data. On the other hand, the very nice agreement of the calculation of the incoherent scattering cross section with the intensity and shape of the bumps reported in the Supplementary Material (Fig. S4) is in our opinion enough for ascribing these bumps to incoherent scattering. Last but not least, optic modes intensity is expected to strongly increase with Q, thus to increase over the Brillouin zone. This is not what we find. As can be observed in Fig. S8 of the Supplementary Material, the bumps are basically q-independent, keeping over the whole Brillouin zone the same shape and intensity.

(9) Between lines 311-314, the authors demonstrated the acoustic nature by a constant normalized intensity over almost all of the Brillouin zone. The logic is not clear.

The concept is explained in detail in the Supplementary Material Appendix C: as can be seen in eq. S1, in the acoustic limit $q \ll G$ (G reciprocal lattice vector), the normalized phonon intensity becomes proportional to the elastic structure factor, i.e. the Bragg peak intensity, and inversely proportional to the Bragg Peak lattice vector. The q dependence is lost, thus it should be constant over q . As we find it constant over all measured q s, then we can confirm that the measured mode has the intensity behaviour of an acoustic mode.

The manuscript is worth being published after addressing the above concerns.

Reviewer #2 (Remarks to the Author):

In the manuscript, Turner et al. report detailed study of phonon dynamics of a high-entropy alloy, Fe₂₀Co₂₀Cr₂₀Mn₂₀Ni₂₀, using inelastic neutron and X-ray scattering. The results demonstrate that phonons in the HEA remain well defined and propagate over the entire Brillouin zone, which is reminiscent of those in binary alloys and simple elements, but opposite of those in glasses and complex metallic alloy. In the light of the fact that HEAs exhibit quite similar thermal transport properties to glasses and CMAs, the present study indicates a different microscopic mechanism in HEAs than in glasses and CMAs.

While it is not the first experimental investigation of the lattice dynamics in the HEA despite that the authors claim it is [see Yang et al., Journal of Materials Science & Technology 99 (2022) 55–60], I think the concrete experimental data, competently executed analysis and new insight into the physics of this system merit publication in Nature Communications.

On the other hand, a major issue should be addressed before acceptance, that is lack of comparison between the HEA and the binary alloys, particularly their thermal transport properties and relationships to phonon dynamics. In the manuscript, the authors present that the phonon linewidth of the HEA is broader than the simple-element metals, which explains why HEAs bear much lower thermal conductivities than the simple-element metals, going from some tens of W/mK down to less than 2 W/mK. They also show that the HEA and binary alloys share similar phonon linewidths. According to the above-mentioned logic, the thermal transport properties of HEA should be in the similar level as that of binary alloys. However, to my knowledge, the thermal conductivity of FeNi alloy is larger than 2 W/mK [G. Ya. Khadzhay et al., Low Temp. Phys. 46, 939 (2020)].

We thank the Referee for his/her comments and for the reference on the published HEA dynamics, of which we were not aware. Still, in yang's work, the $S(q,w)$ is summed on a large Q range, so that it basically gives an information on the GVDOS, and indeed, It is similar to our GVDOS. What we present new here is the measurement of the individual phonons: phonon dispersions and attenuation.

The article cited by the Referee shows that the binary alloy disorder is enough to inhibit the Umklapp peak (or dielectric maximum) and reduce the phonon contribution. Jin shows that the calculated phonon κ are the same between FeNi and FCCMN, so, further chemical disorder is ineffective. Our results explain why: phonon lifetime is not reduced further in this specific HEA. We have added this comment to the manuscript.

Besides, several typos written as “an HEA” are found in both the abstract and main text, which should be corrected as “a HEA”.

We have corrected the typos

Reviewer #3 (Remarks to the Author):

The manuscript by Turner et al. reports a comprehensive experimental study of phonon dynamics in high-entropy alloys (HEAs). The sample is the classic quinary alloy, FeCoCrMnNi, also known as the Cantor alloy. The experiments, based on inelastic neutron scattering and inelastic X-ray scattering measurements, were thorough and covered several temperatures. Both polycrystalline and single-crystal samples were studied. The latter allowed mapping of the phonon dispersion curves and determination of phonon lifetimes, although the precision was kind of poor.

The main conclusion of the paper is that in spite of the complicated chemistry, the phonon behavior of this equiatomic multicomponent alloy is similar to that of Ni and the binary NiFe. This can be seen from the generalized density of states as well as from the phonon dispersion curves mapped with a single-crystal sample. The authors also tried to make point about the phonon peak width which characterizes the phonon life time. The authors demonstrated that the broadening of the phonon peak width could be attributed to fluctuation of force constants. The authors also tried to compare the measured peak widths with calculations by Mu et al., but the quality of data really does not allow any definitive conclusions to be drawn (Fig. 6).

We thank the Referee for his/her reading of the paper.

Still, we don't agree with - and are very surprised by - the Referee comment on the “poor quality” of our data. The quality of the spectra is very good, and it has allowed us to extract with precision the energies and intrinsic widths for both longitudinal and transverse modes.

The comparison with Mu shows that globally there is a good agreement, both in the order of magnitude of the attenuation, which is relevant as we keep the same order of magnitude as in binary alloys, and in the q-dependence, linear for transverse modes, and multi-regime for longitudinal modes. We agree that these remain qualitative assessments, but only with calculations on our HEA we could do a more detailed quantitative comparison.

While the paper reported extensive experimental data on phonons in FeCoCrMnNi HEA, the authors fell short of extracting insights on phonon behavior. For example, why would this alloy behave the same as Ni or NiFe? There are lots of first-principles based calculations, which the authors have alluded to in their reference list. The authors could use these calculations to help interpret their experimental data at the microscopic level. Out of these considerations, I am unable to recommend the paper for Nature Communications. Instead, the paper may be better suited for a more specialized journal.

We don't agree with - and again are extremely surprised by -the Referee 's comment that we don't extract any insight on phonon behaviour! it is really not a founded comment.

We demonstrate here that:

- 1) The FCCMN has the same dispersions as Ni and Fe-Ni, which is the consequence of very similar elastic constants between the single elements, so that the multi-element alloys (binary of 5-elements) have similar elastic constants too.

- 2) The phonon lifetime remains of the same order of magnitude as in FeNi, which is in our opinion unexpected as we have increased the global disorder by increasing the number of components of the alloy, thus the number of possible random environments of a given atomic pair. We explain this result as due to a similar width of the force constant distribution. We have estimated it as 20%, which is indeed what Mu reported for FeNi. The existing theories can explain our results assuming that the force constant fluctuations are not much increased going from 2 to 5 elements. However, calculations on 5 elements alloys have not been done yet and would be needed for explaining our results.

A couple of minor comments/questions

1. I don't quite understand why the authors keep referring to glass. HEA is very different from glass. FeCoCrMnNi is a FCC alloy and by and large behaves like a FCC alloy in many aspects, not just in phonons. In fact, as Fig. 8a shows, there is no Boson Peak in this HEA, which is a ubiquitous feature of glass.

The Referee seems to have missed some recent literature on the theory of the phonon dynamics in glasses, among which the work of Beltukov and al. HEAs have long been looked at as most disordered crystals, and thus halfway between ordered crystals and fully disordered materials. Specifically, they can be thought of as intermediate between a perfect crystalline system, with long and short range order, and a glass, fully disordered, as HEA have an average long range order but a short range disorder.

In glasses it is an unsolved point the role of the different kinds of disorder on glass dynamics and thermal transport properties. Beltukov et al. developed a random matrix model for explaining phonon dynamics in glasses: they used an ordered system with only force constant disorder and could reproduce some glass typical features such as the Boson Peak. The fact that in our HEA we have basically only force constant disorder, allows the comparison with glasses, where such kind of disorder has been pointed at as main responsible for the phonon and thermal anomalies.

2. P. 3, lines 192-196, what is the point of saying "we could see weak signatures of xxx, but this result needs to be confirmed...?"

In HEAs there is much discussion on the presence or not of short range order. We thought it important to mention our preliminary results. However, we agree with the Referee that it is not an essential information, thus we have moved it to the Supplementary Information

3. The massive data sets of phonon scans in Fig. 3 could be condensed or better yet presented in Supplemental Information.

These are only 6 q-points/phonon branch. The rest of the data are reported in the Supplementary Materials. If there is no problem of space, we prefer to keep these in the main text as representative for all data.

4. Fig. 6 and the subsequent discussion are somewhat disconnected. Fig. 6 was first presented in section C, but without much discussion. Discussion of Fig. 6 came after Fig. 7, the comparison with calculations with force constant fluctuations.

Fig. 6 reports our experimental data on the width, thus it is in the results section. We have added in the same figure the data by Mu et al., in order not to re-propose the same figure with only the addition of Mu data for the discussion. However, in the discussion, the logic

order is 1) comparison with a simple perturbation theory 2) comparison with predictions from the most complex theory by Mu. This is why it is commented later.

Reviewers' comments:

Reviewer #1 (Remarks to the Author):

The authors do an excellent job responding to my questions, and the modified manuscript is more logical than before. However, there are still a few concerns:

1) In response to my third question, the authors follow my suggestion on explaining the peak in reduced GVDOS. I am satisfied with this point but still confused with the result of the $\omega_{4.5}$ -to- $\omega_{1.7}$ crossover. In Fig.7, the authors tried to explain the phonon width in HEA with a perturbation theory. The model is straightforward, which matches well with the measurements. One can see two peaks in the lin-lin plot of phonon width to phonon energy. However, in Fig8, the authors tried to display a $\omega_{4.5}$ -to- $\omega_{1.7}$ crossover in a log-log plot of phonon width to phonon energy. The same data is in the same region, but the explanation changed from a peak behavior to a crossover behavior. I am confused by this operation, and I think other readers may also have problems accepting both the peak and crossover behavior.

2) Moreover, due to the anisotropy of crystalline structures, the phonon width and energy may be different in different directions, as can be seen in Fig.7(a) and 7(c). It also means that the $\omega_{4.5}$ -to- $\omega_{1.7}$ crossover may be sensitive to the directions. This is quite different from the case in glass, which is isotropic due to the disordered structures. In my opinion, the crossover behavior in HEA is uncertain and misleading. The authors should be careful with their discussion in this part.

3) It's still confusing what pattern-matching appropriate methods they used. The authors should write it clearer. The difference between measured data and fitted patterns should be given.

4) Generally, the time-of-flight instrument energy resolution decreases to low values as neutron energy loss goes to high. Because slow flight neutrons have better time resolution, I don't understand the statement, "Specifically, it remains less than 1 meV until ~20 meV and is as large as ~2 meV at around 40 meV".

5) The Q integrating range of the GVDOS data shown in Fig.2 should be clarified. It would be suitable for readers if the authors could show the total raw data, the multiphonon scattering, the elastic, and the empty can data in the same GVDOS plot, at least in Supplementary information. So, in that case, readers can know what the raw data look like and how well the background subtraction works.

Reviewer #2 (Remarks to the Author):

The authors have addressed the major issue I pointed out by adding the discussions on the relationships of thermal behavior between binary and multi-element alloy, in particular that the similar phonon contributions (of about 4 and 6.9 W/mk from the references) found in FeNi and FeCoCrMnNi, which is consistent with the discovery in the manuscript that the binary and multi-element alloys share similar phonon linewidths.

Reviewer #3 (Remarks to the Author):

I have read the rebuttal letter carefully. Unfortunately, I do not feel that the authors have taken efforts to address my questions.

Regarding the quality of the data, the authors responded “Still, we don’t agree with - and are very surprised by - the Referee comment on the “poor quality” of our data. The quality of the spectra is very good, and it has allowed us to extract with precision the energies and intrinsic widths for both longitudinal and transverse modes.”

In Fig. 6, we can see that the error bars for phonon linewidth is about 0.5-1 meV. This level of precision, perhaps the best one can achieve from the current INS and IXS data, is inadequate to compare with supercell based first-principles calculations. With this level of precision, I cannot tell whether the experimental phonon width data agree or disagree with the calculations, so the comparison is not very meaningful. On the other hand, for NiFe (data from Mu et al. orange crosses), one can say that the comparison is quite good thus validating the idea of force constant fluctuations due to changes in local chemical environments as proposed by Mu et al. Indeed, it would be helpful to have the calculations precisely for CrMnFeCoNi, in order to make a fair comparison. The authors could have pursued further in this direction if the comparison is important. Körmann et al. (npj Computational Materials (2017) 3:36) have done this kind of calculation for a variety of BCC high entropy alloys (up to five elements) to demonstrate the effect of force constant and mass fluctuations on phonon peak width. Similar calculations could be done for FCC CrMnFeCoNi.

In my early review, I asked the question why the phonons of the quinary Cantor alloy would behave the same as Ni or NiFe. The authors responded that this is because “the multi-element alloys (binary of 5-elements) have similar elastic constants too.” I would like to point out that the five elements (Co, Cr, Ni, Fe, Mn) don’t even have the same crystal structure as a single element material. In fact, only Ni has the FCC structure. Because of differences in crystal symmetry, the elastic behaviors (anisotropy, for example) would be very different. The phonon spectra are also different. In HCP Co, for example, the phonon spectrum is far more complicated than in FCC Co (Wakabayashi, PRB 25, 5122 (1982)).

Regarding the question on the Boson peak (BP), I am glad that the authors have agreed that there is no BP when answering the question from Referee #1. It would be great if the discussion just stopped here. To my surprise, the authors brought out unnecessary discussions on the origin of the BP when answering my question, which is no longer relevant to this paper.

“The Referee seems to have missed some recent literature on the theory of the phonon dynamics in glasses, among which the work of Beltukov and al.... Beltukov et al. developed a random matrix model for explaining phonon dynamics in glasses: they used an ordered system with only force constant disorder and could reproduce some glass typical features such as the Boson Peak.”

It may be helpful to point out there are lots of theories and experimental studies on BP and the nature of the BP is far from being understood. There have been two schools of thought: that the BP comes from force constant fluctuations or the BP could be due to the intrinsic disordered structure of the glassy material. In the first scenario, the BP could be linked to van Hove singularities in transverse phonons as pointed out by Chumakov et al. (PRL, 106, 225501 (2011)). Nevertheless, there is really no point to continue the discussion of BP in this alloy.

In reading the response to other reviewers, I am curious about the following points.

1. No Rietveld refinement was performed for the neutron diffraction pattern because “as the

sample used for neutron diffraction was not powder but polycrystalline with few grains and a non perfect orientational average.” If this is the case, can we expect the VDOS measured with the “powder samples” to be representative, i.e., not influenced by select big grains?

2. “Fig. 2 is a GVDOS obtained integrating at low Q.” What was the Q-range over which the VDOS was measured or summed? In Yang’s paper (JMST, 2022), the summed Q-range was $6 \leq |Q| \leq 10 \text{ \AA}^{-1}$.

I don't think the paper is fit for publication in Nature Communication.

Reviewer #1 (Remarks to the Author):

We thank the Referee for his positive comments to our answers and revised manuscript. Here we answer his last few questions.

The authors do an excellent job responding to my questions, and the modified manuscript is more logical than before. However, there are still a few concerns:

1) In response to my third question, the authors follow my suggestion on explaining the peak in reduced GVDOS. I am satisfied with this point but still confused with the result of the $\omega_{4.5}$ -to- $\omega_{1.7}$ crossover. In Fig.7, the authors tried to explain the phonon width in HEA with a perturbation theory. The model is straightforward, which matches well with the measurements. One can see two peaks in the lin-lin plot of phonon width to phonon energy. However, in Fig8, the authors tried to display a $\omega_{4.5}$ -to- $\omega_{1.7}$ crossover in a log-log plot of phonon width to phonon energy. The same data is in the same region, but the explanation changed from a peak behavior to a crossover behavior. I am confused by this operation, and I think other readers may also have problems accepting both the peak and crossover behavior.

In Fig. 7 two peaks are well visible in the theoretical model. While the agreement is quite good with our data, these latter do not clearly display two peaks, but rather one peak above 30 meV. The good agreement concerns basically the strong increase at low q followed by a weaker increase/almost plateau in the attenuation, and then a maximum above 30 meV. Fig. 8 focuses on the energy region below 30 meV, i.e. before this maximum, where our data do not clearly show a peak but the two regimes. We have specified the limited energy range for the crossover in this new version.

2) Moreover, due to the anisotropy of crystalline structures, the phonon width and energy may be different in different directions, as can be seen in Fig.7(a) and 7(c). It also means that the $\omega_{4.5}$ -to- $\omega_{1.7}$ crossover may be sensitive to the directions. This is quite different from the case in glass, which is isotropic due to the disordered structures. In my opinion, the crossover behavior in HEA is uncertain and misleading. The authors should be careful with their discussion in this part.

Indeed, we comment that this anisotropy comes from the inherent nature of the HEA, as it appears as well in Mu's simulations. The strong-weak regime crossover appears in some complex crystalline materials as well, but not in all directions and polarizations. This is why we say that HEA share with glasses and CMA the presence in some directions and polarizations of this crossover (page 8, line 532 "As such, phonon dynamics in our HEA, while reminiscent in some directions of what is seen in glasses, remains different", page 9 line 603 "Still, all these materials present a strong phonon scattering regime, $\Gamma\text{FWHM} \propto (\hbar\omega)^4$, in at least some polarizations and directions, which arises simultaneously to the deviation of the acoustic dynamics from the Debye behavior.")

3) It's still confusing what pattern-matching appropriate methods they used. The authors should write it clearer. The difference between measured data and fitted patterns should be given.

The pattern-matching is performed with a LeBail fit. This type of refinement means that only the geometrical parameters (sample cell parameters, peak width parameters /experimental wavelength, instrumental peak shape and peak width function including asymmetry, $2\theta_{\text{zero}}$ correction) are accounted for, and that the peak intensities are refined directly against experimental ones, without

accounting for crystal structure. We have explicated it in the main text and added a few more details in the Supplementary Material. We have also modified the figure to include the residuals of the fit as asked by the Referee.

4) Generally, the time-of-flight instrument energy resolution decreases to low values as neutron energy loss goes to high. Because slow flight neutrons have better time resolution, I don't understand the statement, "Specifically, it remains less than 1 meV until ~20 meV and is as large as ~2 meV at around 40 meV".

It seems there is a misunderstanding on our nomenclature. The energy resolution degrades going up in energy, that is why the peak width associated to the instrumental resolution is smaller at low energy and larger at high energy. 1 meV is the resolution-limited peak width until 20 meV, and 2 meV is the resolution-limited peak width at 40 meV: thus the energy resolution is higher at low than high energy.

5) The Q integrating range of the GVDOS data shown in Fig.2 should be clarified. It would be suitable for readers if the authors could show the total raw data, the multiphonon scattering, the elastic, and the empty can data in the same GVDOS plot, at least in Supplementary information. So, in that case, readers can know what the raw data look like and how well the background subtraction works.

The Q range is specified in the Method section. We have now specified it in the caption of the figure as well.

As for showing the total data and the different contributions, this is not trivial concerning the multiphonon scattering, because it is iteratively calculated and subtracted in the MUPHOCOR routine and there is an automatic normalization at each step.

This is the reason of our late answer. We have been working on the MUPHOCOR code in order to be able to extract also this contribution as asked by the Referee.

The figures can now be found in the Supplementary Material, section 2. The first figure shows raw data in time of flight units, together with the empty cell, showing then the empty cell subtraction. As for the elastic line, it appears in the time of flight raw data but not in the GVDOS, as this latter is proportional to $\omega S(\omega)$, giving zero at zero energy. The second figure reports the GVDOS before and after the multiphonon subtraction. The multiphonon contribution, calculated with the self consistent iterative procedure, is also reported in the figure.

Reviewer #2 (Remarks to the Author):

We thank the Referee for his/her comments.

The authors have addressed the major issue I pointed out by adding the discussions on the relationships of thermal behavior between binary and multi-element alloy, in particular that the similar phonon contributions (of about 4 and 6.9 W/mK from the references) found in FeNi and FeCoCrMnNi, which is consistent with the discovery in the manuscript that the binary and multi-element alloys share similar phonon linewidths.

Reviewer #3 (Remarks to the Author):

I have read the rebuttal letter carefully. Unfortunately, I do not feel that the authors have taken efforts to address my questions.

Regarding the quality of the data, the authors responded “Still, we don’t agree with - and are very surprised by - the Referee comment on the “poor quality” of our data. The quality of the spectra is very good, and it has allowed us to extract with precision the energies and intrinsic widths for both longitudinal and transverse modes.”

In Fig. 6, we can see that the error bars for phonon linewidth is about 0.5-1 meV. This level of precision, perhaps the best one can achieve from the current INS and IXS data, is inadequate to compare with supercell based first-principles calculations. With this level of precision, I cannot tell whether the experimental phonon width data agree or disagree with the calculations, so the comparison is not very meaningful. On the other hand, for NiFe (data from Mu et al. orange crosses), one can say that the comparison is quite good thus validating the idea of force constant fluctuations due to changes in local chemical environments as proposed by Mu et al.

Indeed our data have errorbars which are at the state of the art for this kind of data as measured by inelastic x ray scattering with our energy resolution. We note that the data on NiFe from Mu et al. have similar and even larger errorbars, as can be appreciated from their fig. 3, reported here as well:

Those data have been used by Mu to compare with his supercell based first-principles calculations and comment on the very good agreement. As such, our data can be similarly compared with Mu’s results.

Indeed, it would be helpful to have the calculations precisely for CrMnFeCoNi, in order to make a fair comparison. The authors could have pursued further in this direction if the comparison is important. Körmann et al. (npj Computational Materials (2017) 3:36) have done this kind of calculation for a variety of BCC high entropy alloys (up to five elements) to demonstrate the effect of force constant and mass fluctuations on phonon peak width. Similar calculations could be done for FCC CrMnFeCoNi.

We agree with the Referee that it would be great to compare with calculations on the very same system. Still, as the Referee says, Körmann only calculated the phonon dynamics in BCC HEAs, and there is still work to do on FCC. On the other hand, the Körmann paper is purely theoretical and

misses a comparison with experiments. Our work is experimental, and meant to stimulate calculations on FCC HEAs with 5 elements.

In my early review, I asked the question why the phonons of the quinary Canton alloy would behave the same as Ni or NiFe. The authors responded that this is because “the multi-element alloys (binary of 5-elements) have similar elastic constants too.” I would like to point out that the five elements (Co, Cr, Ni, Fe, Mn) don’t even have the same crystal structure as a single element material. In fact, only Ni has the FCC structure. Because of differences in crystal symmetry, the elastic behaviors (anisotropy, for example) would be very different. The phonon spectra are also different. In HCP Co, for example, the phonon spectrum is far more complicated than in FCC Co (Wakabayashi, PRB 25, 5122 (1982)).

It is true that we may expect differences between our quinary alloy and the elements and binary alloys of the same elements, due to the differences among the elements. Still, we find that at small q the agreement is very good, and this is true also for other binary alloys of the same elements (Fe-Mn [Y. Endoh, Y. Noda, and M. Iizumi, J. Phys. Soc. Jpn. 50, 469 (1981).], Fe-Co [S. M. Shapiro and S. C. Moss, Phys. Rev. B 15, 2726(1977).], and Co-Ni [S. Mu, et al. npj Computational Materials 6, 4 (2020)., F. Menzinger, F. Sacchetti, and M. C. Spinelli, Phys. Rev. B 12, 2253 (1975)]), as we report here for the Referee. Such similarity means that the average elastic constants are very similar. On the other hand, Mu calculated the force constant distribution for Fe-Ni, Fe-Co and Fe-Ni-Co, and found that, while the width of the distribution was significantly different, the center of mass of the distribution was quite similar from one system to the other (Fig.5 of Mu’s paper). As the Referee points out, this is not necessarily expected: it is an original experimental result.

Regarding the question on the Boson peak (BP), I am glad that the authors have agreed that there is no BP when answering the question from Referee #1. It would be great if the discussion just stopped

here. To my surprise, the authors brought out unnecessary discussions on the origin of the BP when answering my question, which is no longer relevant to this paper.

“The Referee seems to have missed some recent literature on the theory of the phonon dynamics in glasses, among which the work of Beltukov and al.... Beltukov et al. developed a random matrix model for explaining phonon dynamics in glasses: they used an ordered system with only force constant disorder and could reproduce some glass typical features such as the Boson Peak.”

It may be helpful to point out there are lots of theories and experimental studies on BP and the nature of the BP is far from being understood. There have been two schools of thought: that the BP comes from force constant fluctuations or the BP could be due to the intrinsic disordered structure of the glassy material. In the first scenario, the BP could be linked to van Hove singularities in transverse phonons as pointed out by Chumakov et al. (PRL, 106, 225501 (2011)). Nevertheless, there is really no point to continue the discussion of BP in this alloy.

We agree with the Referee that the discussion on the Boson Peak origin is out of the scope of our article, and indeed it is not in the article. The Referee does not appreciate our answer, where we explain the relevance of the comparison between HEA and glasses. In order to answer the referee, we have just recalled that in a random matrix model Beltukov could find some glassy features such as the Boson Peak and the strong scattering regime. Being the random matrix model a very good representation of our material, it is thus relevant to compare it to glasses. Moreover, given the different schools of thought on the origin of the BP, our results are interesting, as they show that relatively weak elastic heterogeneities are not enough for generating the BP.

In reading the response to other reviewers, I am curious about the following points.

1. No Rietveld refinement was performed for the neutron diffraction pattern because “as the sample used for neutron diffraction was not powder but polycrystalline with few grains and a non perfect orientational average.” If this is the case, can we expect the VDOS measured with the “powder samples” to be representative, i.e., not influenced by select big grains?

The neutron diffraction experiment is in a theta-2theta configuration, in a 1D geometry. As such it cannot average over the reciprocal space. The GVDOS is integrated in a large Q range, both in modulus and direction, through a 2D detector, thus the results are not affected by the fact that the sample was polycrystalline and not a perfect powder.

2. “Fig. 2 is a GVDOS obtained integrating at low Q.” What was the Q-range over which the VDOS was measured or summed? In Yang’s paper (JMST, 2022), the summed Q-range was $6 \leq |Q| \leq 10 \text{ \AA}^{-1}$.

The Q range is specified in the Methods section. In this last version we specify it also in the caption of the figure

I don't think the paper is fit for publication in Nature Communication.

REVIEWERS' COMMENTS

Reviewer #1 (Remarks to the Author):

The authors have partly addressed my concerns, but I am still confused with the discussion on the $\omega_{4.5}$ -to- $\omega_{1.7}$ crossover behavior of phonon width. The authors convince us first that binary and high-entropy alloys share similar phonon GDOS and phonon width. Then, they try to convince us that glasses and high-entropy alloys share similar crossover behavior of phonon widths. This paper is publishable subject to minor revisions noted.

In fact, even in glassy materials, it is still unclear whether the ω_4 -to- ω_2 crossover is a general feature or not. Although in some glasses like vitreous silica (ref62), IXS experiments observed such a ω_4 -to- ω_2 crossover behavior, in metallic glasses (MGs) the IXS experiments observed a pretty different ω_2 -to- ω_1 crossover behavior. For example, Ruocco and coworkers marked a transition from the Q^2 to a linear dependence of the phonon width around $Q=7$ nm⁻¹ in Ni₆₇Zr₃₃ MG [T. Scopigno, J. B. Suck, R. Angelini, F. Albergamo, and G. Ruocco, High-Frequency Dynamics in Metallic Glasses, Phys. Rev. Lett. 96, 135501 (2006)]. Whereas Monaco and coworkers observed a similar transition of the phonon width around $Q=3$ nm⁻¹ in Pd₇₇Si_{16.5}Cu_{6.5} MG (ref72). Therefore, the phonon width may change differently in different glassy materials. In MGs, the phonon width may have a pretty different ω_2 -to- ω_1 crossover behavior. So, the high-entropy alloys and MGs may have quite different phonon width behavior. In my view, the discussion on the similarity between high-entropy alloys and glasses is redundant and misleading, which could be removed away.

R#1's comment to the report of R#3

The authors have addressed all the questions I pointed out, and I am satisfied with the revised manuscript. I also think that the authors have done an excellent job in answering the questions by Reviewer #3. Actually, I think the quality of the data in this paper is good enough to get the main conclusions. Also, there is no need to do similar calculations for the FeCoCrMnNi alloy. As for the new observation that the phonons of the high entropy alloy would behave the same as Ni or NiFe, the authors have also provided a possible understanding. Of course, there could be another explanation, which must remain consistent with this paper's new experimental observation. The experimental results reported in this paper bring about significant advances in studying phonon dynamics and disordered alloys. It will further stimulate broad interest in the physics of complex systems and materials science. Therefore, I suggest the paper be published in NCOMMS.

Reviewer #2 (Remarks to the Author):

R#2's comment to the report of R#3

I went through the comments and rebuttal, share with you my personal opinions on each remarks by Reviewer #3.

"1. On comparison with DFT calculations for the CrMnFeCoNi quinary alloy. I was hoping the authors could present some DFT calculations (of their own or from literature) to compare with, but there were none in the rebuttal letter or the revised manuscript."

My opinion: Adding some DFT calculations on the exact system definitely would be a plus to

the conclusion. However, tremendous work by a specialist is necessary in order to obtain a reliable DFT result, which could be worth of formulating another paper. In the light of the present manuscript features on the experimental discoveries, I think it is an excessive request that the authors add DFT calculations in the paper.

2. On my comments "In my early review, ... (1982)." The authors did not address my comments and simply went back to the argument of similar elastic constants. I don't agree with this argument, as explained in my earlier review.

My opinion: I agree with the reviewer that the authors did not address the issue why the phonons of the quinary Canton alloy would behave the same as Ni in the light of different crystal symmetries.

3. The discussion of the Boson peak is speculative and should be minimized. As mentioned earlier, I don't really see the Boson Peak in Fig. 8a. The GVDOS/E² is flat and is approximately 1. Where is the Boson Peak? In a glass, the Boson Peak is usually related to the Ioffe-Regel crossover where the phonon peak width catches up with the Boson Energy, see Shintani and Tanaka (Nature Materials, 7, 842 (2008)) and references therein. The IR crossover in this HEA is ~30 meV, as the authors showed in Fig. 8c, far above the normal range for the Boson Peak. So I don't see the connection with the Boson Peak here.

My opinion: Both sides agrees that Boson peak is not essential in the manuscript.

Regarding the answer to my minor question #2, the Q-range indicated by the authors is small (0-2 Å⁻¹ and 0-3.6 Å⁻¹), compared to the normal range of integration for time-of-flight instruments (e.g., 6-10 Å⁻¹ in the reference mentioned by Referee #2, JMST 2022) where the phonon scattering dominates. For HEAs, there is a further complication by the magnetic scattering at low-Q, which the authors said was quite strong. That being case, I have questions about the accuracy of the GVDOS in Fig. 2.

My opinion: This is a minor question from the reviewer. I think the authors can add some discussion about the magnetic scattering contribution at low-Q region to address it.

Reviewer #3 (Remarks to the Author):

I appreciate the authors making efforts to answer my questions. After carefully reading through the response letter and the revised manuscript, however, I found that the authors did not really address the core of my questions. My fundamental question was about the insights that one can extract from experimental data, and I raised the question from several angles. The authors mostly circle around my questions and spent a lot of time arguing based on evidence or arguments already in the previous versions of the manuscript.

1. On comparison with DFT calculations for the CrMnFeCoNi quinary alloy. I was hoping the authors could present some DFT calculations (of their own or from literature) to compare with, but there were none in the rebuttal letter or the revised manuscript.

On a minor point, it would have been nice if the authors labeled figures in the rebuttal letter and included a caption and source, rather than for the reviewers to figure out what these figures are.

2. On my comments "In my early review, ... (1982)." The authors did not address my

comments and simply went back to the argument of similar elastic constants. I don't agree with this argument, as explained in my earlier review.

Once again, the figure in this section was unlabeled and contained no caption.

3. The discussion of the Boson peak is speculative and should be minimized. As mentioned earlier, I don't really see the Boson Peak in Fig. 8a. The GVDOS/E² is flat and is approximately 1. Where is the Boson Peak? In a glass, the Boson Peak is usually related to the Ioffe-Regel crossover where the phonon peak width catches up with the Boson Energy, see Shintani and Tanaka (Nature Materials, 7, 842 (2008)) and references therein. The IR crossover in this HEA is ~30 meV, as the authors showed in Fig. 8c, far above the normal range for the Boson Peak. So I don't see the connection with the Boson Peak here.

Regarding the answer to my minor question #2, the Q-range indicated by the authors is small (0-2 Å⁻¹ and 0-3.6 Å⁻¹), compared to the normal range of integration for time-of-flight instruments (e.g., 6-10 Å⁻¹ in the reference mentioned by Referee #2, JMST 2022) where the phonon scattering dominates. For HEAs, there is a further complication by the magnetic scattering at low-Q, which the authors said was quite strong. That being the case, I have questions about the accuracy of the GVDOS in Fig. 2.

Overall, the authors haven't answered my questions, and I don't think this manuscript has changed much from the previous versions.

REVIEWERS' COMMENTS

Reviewer #1 (Remarks to the Author):

The authors have partly addressed my concerns, but I am still confused with the discussion on the $\omega_{4.5}$ -to- $\omega_{1.7}$ crossover behavior of phonon width. The authors convince us first that binary and high-entropy alloys share similar phonon GDOS and phonon width. Then, they try to convince us that glasses and high-entropy alloys share similar crossover behavior of phonon widths. This paper is publishable subject to minor revisions noted.

In fact, even in glassy materials, it is still unclear whether the ω_4 -to- ω_2 crossover is a general feature or not. Although in some glasses like vitreous silica (ref62), IXS experiments observed such a ω_4 -to- ω_2 crossover behavior, in metallic glasses (MGs) the IXS experiments observed a pretty different ω_2 -to- ω_1 crossover behavior. For example, Ruocco and coworkers marked a transition from the Q² to a linear dependence of the phonon width around Q=7 nm⁻¹ in Ni₆₇Zr₃₃ MG [T. Scopigno, J. B. Suck, R. Angelini, F. Albergamo, and G. Ruocco, High-Frequency Dynamics in Metallic Glasses, Phys. Rev. Lett. 96, 135501 (2006).]. Whereas Monaco and coworkers observed a similar transition of the phonon width around Q=3 nm⁻¹ in Pd₇₇Si_{16.5}Cu_{6.5} MG (ref72). Therefore, the phonon width may change differently in different glassy materials. In MGs, the phonon width may have a pretty different ω_2 -to- ω_1 crossover behavior. So, the high-entropy alloys and MGs may have quite different phonon width behavior. In my view, the discussion on the similarity between high-entropy alloys and glasses is redundant and misleading, which could be removed away.

We thank the Referee for his/her critical view and suggestions. We have preferred not to remove the comparison with glasses, which is motivated by the will to understand the role of force constant disorder over the topological or mass disorder, as explained in the introduction. We have however rewritten this comparison, reminding to the reader the reason for it, and moderating it, taking into account all the comments of the Referee and reducing the extent of this part of the discussion.

R#1's comment to the report of R#3

The authors have addressed all the questions I pointed out, and I am satisfied with the revised manuscript. I also think that the authors have done an excellent job in answering the questions by Reviewer #3. Actually, I think the quality of the data in this paper is good enough to get the main conclusions. Also, there is no need to do similar calculations for the FeCoCrMnNi alloy. As for the new observation that the phonons of the high entropy alloy would behave the same as Ni or NiFe, the authors have also provided a possible understanding. Of course, there could be another explanation, which must remain consistent with this paper's new experimental observation. The experimental results reported in this paper bring about significant advances in studying phonon dynamics and disordered alloys. It will further stimulate broad interest in the physics of complex systems and materials science. Therefore, I suggest the paper be published in NCOMMS.

Reviewer #2 (Remarks to the Author):

We thank the Referee for his/her comments on the report of R#3

R#2's comment to the report of R#3

I went through the comments and rebuttal, share with you my personal opinions on each remarks by Reviewer #3.

"1. On comparison with DFT calculations for the CrMnFeCoNi quinary alloy. I was hoping the authors could present some DFT calculations (of their own or from literature) to compare with, but there were none in the rebuttal letter or the revised manuscript."

My opinion: Adding some DFT calculations on the exact system definitely would be a plus to the conclusion. However, tremendous work by a specialist is necessary in order to obtain a reliable DFT result, which could be worth of formulating another paper. In the light of the present manuscript features on the experimental discoveries, I think it is an excessive request that the authors add DFT calculations in the paper.

2. On my comments "In my early review, ... (1982)." The authors did not address my comments and simply went back to the argument of similar elastic constants. I don't agree with this argument, as explained in my earlier review.

My opinion: I agree with the reviewer that the authors did not address the issue why the phonons of the quinary Cantor alloy would behave the same as Ni in the light of different crystal symmetries.

With reference to this question: if we plot the acoustic dispersions in absolute units, rather than relative units, their slope is the sound velocity, which is related to the elastic constant divided by the density. Comparing sound velocities of the elements, we find that they are quite similar with a standard deviation of 9%. This explains the good agreement of the slope of our Cantor alloy with the ones of the single elements, as, independently on the crystallographic structure, sound velocities are very close to each other. We have corrected the manuscript pointing to the similarity of the sound velocities rather than the elastic constants.

3. The discussion of the Boson peak is speculative and should be minimized. As mentioned earlier, I don't really see the Boson Peak in Fig. 8a. The GVDOS/E² is flat and is approximately 1. Where is the Boson Peak? In a glass, the Boson Peak is usually related to the Ioffe-Regel crossover where the phonon peak width catches up with the Boson Energy, see Shintani and Tanaka (Nature Materials, 7, 842 (2008)) and references therein. The IR crossover in this HEA is ~30 meV, as the authors showed in Fig. 8c, far above the normal range for the Boson Peak. So I don't see the connection with the Boson Peak here.

My opinion: Both sides agrees that Boson peak is not essential in the manuscript.

We have also reduced the discussion on the Boson Peak in this last revision.

Regarding the answer to my minor question #2, the Q-range indicated by the authors is small (0-2 Å⁻¹ and 0-3.6 Å⁻¹), compared to the normal range of integration for time-of-flight instruments (e.g., 6-10 Å⁻¹ in the reference mentioned by Referee #2, JMST 2022) where the phonon scattering dominates. For HEAs, there is a further complication by the magnetic scattering at low-Q, which the authors said was quite strong. That being case, I have questions about the accuracy of the GVDOS in Fig. 2.

My opinion: This is a minor question from the reviewer. I think the authors can add some discussion about the magnetic scattering contribution at low-Q region to address it.

We have added in the text some discussion, specifically indicating that the larger Q range allows to reduce by a factor of 75-80% the magnetic contribution to the GVDOS, and the recovery of the E^2 behaviour confirms that the magnetic contribution has been minimized

Reviewer #3 (Remarks to the Author):

I appreciate the authors making efforts to answer my questions. After carefully reading through the response letter and the revised manuscript, however, I found that the authors did not really address the core of my questions. My fundamental question was about the insights that one can extract from experimental data, and I raised the question from several angles. The authors mostly circle around my questions and spent a lot of time arguing based on evidence or arguments already in the previous versions of the manuscript.

1. On comparison with DFT calculations for the CrMnFeCoNi quinary alloy. I was hoping the authors could present some DFT calculations (of their own or from literature) to compare with, but there were none in the rebuttal letter or the revised manuscript.

On a minor point, it would have been nice if the authors labeled figures in the rebuttal letter and included a caption and source, rather than for the reviewers to figure out what these figures are.

2. On my comments "In my early review, ... (1982)." The authors did not address my comments and simply went back to the argument of similar elastic constants. I don't agree with this argument, as explained in my earlier review.

Once again, the figure in this section was unlabeled and contained no caption.

If we plot the acoustic dispersions in absolute units, rather than relative units, their slope is the sound velocity, which is related to the elastic constant divided by the density. Comparing sound velocities of the elements, we find that they are quite similar with a standard deviation of 9%. This explains the good agreement of the slope of our Cantor alloy with the ones of the single elements, as, independently on the crystallographic structure, sound velocities are very close to each other. We have corrected the manuscript pointing to the similarity of the sound velocities rather than the elastic constants.

3. The discussion of the Boson peak is speculative and should be minimized. As mentioned earlier, I don't really see the Boson Peak in Fig. 8a. The GVDOS/ E^2 is flat and is approximately 1. Where is the Boson Peak? In a glass, the Boson Peak is usually related to the Ioffe-Regel crossover where the phonon peak width catches up with the Boson Energy, see Shintani and Tanaka (Nature Materials, 7, 842 (2008)) and references therein. The IR crossover in this HEA is ~ 30 meV, as the authors showed in Fig. 8c, far above the normal range for the Boson Peak. So I don't see the connection with the Boson Peak here.

We have reduced the discussion on the Boson Peak in this last revision.

Regarding the answer to my minor question #2, the Q-range indicated by the authors is small ($0-2 \text{ \AA}^{-1}$ and $0-3.6 \text{ \AA}^{-1}$), compared to the normal range of integration for time-of-flight instruments (e.g., $6-10 \text{ \AA}^{-1}$ in the reference mentioned by Referee #2, JMST 2022) where the phonon scattering dominates. For HEAs, there is a further complication by the magnetic scattering at low-Q, which the

authors said was quite strong. That being case, I have questions about the accuracy of the GVDOS in Fig. 2.

We have added in the text some discussion, specifically indicating that the larger Q range allows to reduce by a factor of 75-80% the magnetic contribution to the GVDOS, and the recovery of the E^2 behaviour confirms that the magnetic contribution has been minimized

Overall, the authors haven't answered my questions, and I don't think this manuscript has changed much from the previous versions.